# Who's Gaming the System? A Causally-Motivated Approach for Detecting Strategic Adaptation

**Trenton Chang**[1]    **Lindsay Warrenburg**[2]    **Sae-Hwan Park**[2]    **Ravi B. Parikh**[2,3]
**Maggie Makar**[1]    **Jenna Wiens**[1]
[1]University of Michigan    [2]University of Pennsylvania    [3]Emory University
`{ctrenton,mmakar,wiensj}@umich.edu`
`{lindsay.warrenburg,sae-hwan.park}@pennmedicine.upenn.edu`
`ravi.bharat.parikh@emory.edu`

## Abstract

In many settings, machine learning models may be used to inform decisions that impact individuals or entities who interact with the model. Such entities, or *agents*, may *game* model decisions by manipulating their inputs to the model to obtain better outcomes and maximize some utility. We consider a multi-agent setting where the goal is to identify the "worst offenders:" agents that are gaming most aggressively. However, identifying such agents is difficult without being able to evaluate their utility function. Thus, we introduce a framework featuring a gaming deterrence parameter, a scalar that quantifies an agent's (un)willingness to game. We show that this gaming parameter is only partially identifiable. By recasting the problem as a causal effect estimation problem where different agents represent different "treatments," we prove that a ranking of all agents by their gaming parameters is identifiable. We present empirical results in a synthetic data study validating the usage of causal effect estimation for gaming detection and show in a case study of diagnosis coding behavior in the U.S. that our approach highlights features associated with gaming.

## 1 Introduction

Machine learning (ML) models often guide decisions that impact individuals or entities. Attributes describing an individual or entity are often inputs to such models. In response, such entities may modify their attributes to obtain a more desirable outcome. But changing one's attributes may be costly due to the difficulty of generating supporting evidence, or penalties for fraud. This behavior is called *gaming* or *strategic adaptation* [1]. Strategic adaptation frames gaming as "utility maximization:" *agents* change their attributes to maximize a payout, but incur a cost for modifying attributes.

As an illustrative example, we turn to the health insurance industry. In the United States (U.S.), contracted health insurance companies report their enrollees' diagnoses to the government, which calculates a payout based on reported diagnoses via a publicly available model [2]. The *payout* is intended to support care of the enrollee in relation to the diagnosis. Companies may attempt to maximize payouts by reporting extraneous diagnoses, an illegal practice known as "upcoding" [3]. Despite increasing awareness of upcoding [3, 4, 5, 6], upcoding costs U.S. taxpayers over \$12B U.S. dollars annually [7], even with substantial investment in audits (\$100.7M U.S. dollars, 2023 [8]) and payout changes to adjust for gaming [9, 10]. Since audits may not scale and often overlook fraud [11, 12], tools for flagging gaming-prone agents could help target audits. Beyond health insurance, gaming emerges in responses to credit-scoring algorithms [13, 14] and driver responses to rider allocation algorithms in ride-sharing apps [15].

38th Conference on Neural Information Processing Systems (NeurIPS 2024).

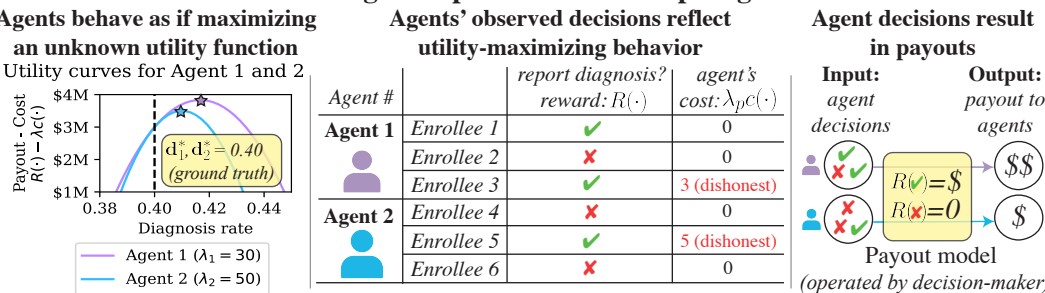

Figure 1: **Left:** Two agents with gaming deterrence parameters $\lambda_1 = 30$ (purple) and $\lambda_2 = 50$ (blue) maximize utility (reward $R$ - cost $c$) with respect to diagnosis rate. Gaming costs *increase* in $\lambda_{(\cdot)}$, and lower an agent's optimal diagnosis rate (stars). **Center:** Agents' observed decisions reflect utility-maximizing behavior. **Right:** A decision-maker computes a payout based on agent decisions.

In this work, we study how one can identify agents with the highest propensity to strategically manipulate their inputs given a dataset of agents and their observed model inputs. A supervised approach is infeasible since fraud/gaming labels are unavailable in our setting. A common paradigm for fraud/gaming detection is unsupervised anomaly detection. However, gamed attributes may not be outlier-like. The perspective of gaming as utility-maximizing behavior in strategic classification provides an alternative to existing fraud/gaming detection methods. Many works in strategic classification (*e.g.*, [1]) assume known utility functions and identical feature manipulation costs across agents, which assists in identifying an agent's "optimal" gaming behavior. However, such assumptions may not always apply. For example, in U.S. Medicare, due to the rarity of penalties for upcoding, it is unclear how to quantify the *cost* of fraud. Furthermore, due to the large number of companies contracted with U.S. Medicare, with distinct incentives (*e.g.*, for-profit vs. non-profit groups) and disjoint populations of patients, there may be heterogeneity in feature manipulation costs.

To bridge this gap, we propose a novel framework for modeling agent utilities by introducing a **gaming deterrence parameter**, which scales the perceived cost (to the agent) of gaming. First, we show that directly estimating the gaming parameter is not possible: the best we can do is a lower bound on the gaming deterrence parameter. However, by re-casting gaming detection as a causal effect estimation problem, where each agent represents a "treatment," we prove that a *ranking* of agents based on their gaming deterrence parameter is recoverable. Thus, we propose a causally-motivated ranking algorithm that produces a ranking of agents. Practically, agents most likely to game under our ranking could be flagged for further monitoring/auditing. While our framework is inspired by health insurance fraud, it applies more broadly to instances of gaming where multiple agents are gaming an ML-guided decision.

We evaluate the performance of causal effect estimators for gaming detection in a synthetic dataset. Empirically, causal approaches rank the worst offenders higher than existing non-causal approaches that screen based on payouts/randomly, as well as anomaly detection methods. We then verify in a real-world U.S. Medicare claims dataset that causal effect estimation yields rankings correlated with the prevalence of for-profit healthcare providers, a suspected driver of gaming [4, 16].

In summary: we **1)** extend strategic classification to model differences in gaming behavior across agents (Section 3), **2)** prove that point-identifying an agents' gaming parameter is impossible without strong assumptions (Section 4), **3)** show that, by recasting gaming detection as causal effect estimation, one can recover a ranking of agents based on their gaming deterrence parameters (Section 4), **4)** demonstrate empirically that our framework identifies the worst offenders with fewer audits than baselines (Section 5), and **5)** show in a real-data case study that our approach yields rankings correlated with suspected drivers of gaming (Section 5). Code to replicate our experiments will be made publicly available at `https://github.com/MLD3/gaming_detection`.

## 2 Related works

**Machine learning-based anomaly detection.** Fraud detection is often framed as an unsupervised anomaly/outlier detection problem [17, 18, 19, 20, 21, 22, 23]. Such approaches assume gamed model inputs are outliers in some distribution. However, this assumption may be incorrect if gaming

is common in the data, or if gaming results in only small changes to observed agent attributes. In borrowing from strategic adaptation, we frame gaming as utility maximization, rather than making distributional assumptions about gamed agent attributes.

**Strategic classification & gaming in machine learning.** A large body of work in strategic classification aims to design incentives to mitigate gaming/strategic behavior by agents. However, such works assume that feature manipulations costs are known/can be estimated across agents, or are identical [1, 24, 25, 26, 27, 28, 29, 30, 31]. In contrast, in our setting, feature manipulation costs are unknown and may differ across agents, but exhibit some shared structure that facilitates comparisons across agents. Shao et al. [32] assumes that agent gaming capacities may differ by placing bounds on manipulation, which may be unrealistic. Closest to our work is that of Dong et al. [33], which also assumes unknown and differing agent costs, but does not leverage similarities in gaming across agents. Our work supplements the strategic classification literature by studying a related yet fundamentally distinct problem: rather than designing incentives to mitigate strategic behavior, which is infeasible under unknown manipulation costs, we leverage differences in costs across agents to identify agents more likely to game.

## 3 Background & Problem Setup

We review strategic adaptation and extend it to model differences in gaming across agents.

**What is strategic adaptation?** Our work builds upon strategic classification [1]. Consider a pre-existing model $f : \mathcal{D} \mapsto \mathbb{R}$ that maps agent attributes $d \in \mathcal{D}$ to a payout. An *agent* may leverage its knowledge of $f$ to change their attribute(s) $d$ according to some function $\Delta$:

$$\Delta(d) \triangleq \underset{\tilde{d} \in \mathcal{D}}{\arg\max}\ R(\tilde{d}; f) - g(\tilde{d}, d) \tag{1}$$

where $R : \mathcal{D} \to \mathbb{R}$ is the *payout* of changing $d$ to $\tilde{d}$, and $g : \mathcal{D} \times \mathcal{D} \to \mathbb{R}_+$ is the *cost* of manipulating $d$. When $\mathcal{D} \subseteq \mathbb{R}$, $g$ is often assumed to be "separable;" *i.e.*, for some function $c$, $g(\tilde{d}, d) = c(\tilde{d} - d)$. $\Delta(\cdot)$ describes how an agent manipulates $d$ to obtain a higher payout from $f$. This behavior is called *strategic adaptation* or *gaming*. For simplicity, we assume $R = f$; *i.e.*, the model $f$ directly determines the payout.

**Modeling agent variation in gaming.** To extend strategic adaptation to multiple agents, we add a non-negative *gaming deterrence parameter* $\lambda_p \in \mathbb{R}^+$ to Eq. 1. Consider an observational dataset $\mathcal{D}_p \triangleq \{(\mathbf{x}_i, d_i)\}_{i=1}^{M_p}$ of an agent $p$'s decisions $d_i \in \{0, 1\}$ given some information $\mathbf{x}_i \in \mathcal{X}$. For simplicity, we assume $d_i$ is binary, though the proposed framework generalizes to non-binary decisions and arbitrary numbers of independent decisions. For example, a health insurance plan $p$ chooses whether to report that an enrollee has a diagnosis $d_i$ given enrollee characteristics $\mathbf{x}_i$. Agent assignment is *mutually exclusive* (*e.g.*, individuals are enrolled in one health insurance plan).

If the agent knows that $d_i$ will be used as input to a payout model, they may have an incentive to *increase* $d_i$ (without loss of generality) to obtain a higher payout. Let $\bar{d} = \frac{1}{M_p} \sum_{i=1}^{M_p} d_i$, and suppose each agent $p$ chooses $\bar{d}$ according to the following utility-maximization problem:

$$P(d_i = 1 \mid p) \equiv \Delta_p(d_p^*) \triangleq \underset{\bar{d} \in [0,1]}{\arg\max}\ R(\bar{d}) - \lambda_p c(\bar{d} - d_p^*) \tag{2}$$

where $R : [0, 1] \to \mathbb{R}$, $c : \mathbb{R} \to \mathbb{R}_+$, and $d_p^*$ is the ground truth value of $\bar{d}$ given the $\mathbf{x}_i$ seen by agent $p$. Thus, $\Delta_p(d_p^*)$ is the gamed/observed decision rate of agent $p$ in a population where the ground truth decision rate is $d_p^*$. Although $\bar{d} \in [0, 1]$ since it is a proportion, our framework applies to $\bar{d}$ on arbitrary intervals. This formulation assumes that each $\mathbf{x}_i$ is equally likely to be gamed, that $\mathbf{x}_i$ are truthfully observed, and that any difference between $\Delta_p(d_p^*)$ and $d_p^*$ is due to gaming.

We focus on the gaming deterrence parameter $\lambda_p$, which scales the cost of manipulation $c(\cdot)$. $\lambda_p$ is non-negative and represents an agent's "aversion" to gaming. Lower values of $\lambda_p$ mean that agent $p$ is more willing to game. Thus, identifying agents most willing to game means finding agents with the lowest $\lambda_p$. We summarize multi-agent strategic adaptation in Figure 1. Next, we introduce assumptions on the reward function $R$, cost function $c$, and ground truth $d_p^*$.

**Assumption 1** (Shared rewards & costs). *Reward ($R$) and cost ($c$) functions are shared across agents.*

Sharing $R$ encodes the assumption that all agents are reacting to the same payout model, while sharing $c$ across agents encodes the belief agents must take similarly-costly actions to manipulate features (*e.g.*, if insurance plans must follow/fraudulently report specific procedures to justify a diagnosis). Many works in strategic classification implicitly make a similar assumption (*e.g.*, [1, 34]).

**Assumption 2** (Increasing rewards). *The reward function $R$ is strictly increasing in $\bar{d}$.*

Increasing rewards formalizes the agent's incentive to perturb its decisions (*i.e.*, inputs to the payout model) from $d_i = 0$ to 1 in Eq. 2.

**Assumption 3** (Cost convexity). *The cost function $c$ is strictly convex and minimized at 0 such that $c(0) = 0$ and $c'(0) = 0$, and increases for all agents $p$ for any $\bar{d} \geq d_p^*$.*

One possible $c$ is $c(x) = x^2$. Strict convexity ensures a unique cost-minimizing action, and $c(0) = 0$ ensures that $d_p^*$ is ground truth, such that increasing $\bar{d}$ incurs greater cost (*e.g.*, Fig. 1, left).

**Assumption 4** (Diminishing or linear returns). *The reward function $R$ is concave in $d_i$.*

For example, $R$ may be a $\log$ or affine function. Assumption 3 (strictly convex $c$) and 4 ensure that $R$ cannot grow fast enough to offset manipulation costs. Furthermore, due to Assumptions 2- 4:

**Remark 1** (Gaming is utility-maximizing). *Given any agent $p$ and $d_p^*$, we have that $\Delta_p(d_p^*) \geq d_p^*$.*

Equivalently, optimal gaming entails increasing $d_i$ from the ground truth. Note that Assumptions 2- 4 are more general versions of assumptions placed on rewards/costs in the strategic classification literature (*e.g.*, [1, 33, 25]).

**Assumption 5** (Non-strategic behavior is feasible). *$d_p^* \in [0, 1]$ is a constant depending solely on $\mathbf{x}_i$.*

Due to Assumption 5, ground truth $d_p^*$ may vary by agent due to differences in $\mathbf{x}_i$ (*e.g.*, health insurance plans serve populations with varying levels of health).

## 4 Theoretical analysis: finding agents most likely to game

We aim to identify agents most likely to game a decision-making model, *i.e.*, agents with the lowest gaming parameters $\lambda_p$. Here, we prove that $\lambda_p$ cannot be point-identified without further assumptions (Section 4.1), but ranking $\lambda_p$ is possible via causal effect estimation (Section 4.2). Detailed proofs are in Appendix B.

### 4.1 Partial identification of the gaming parameter

Here, we show that given our assumptions, $\lambda_p$ is only partially identifiable (cannot be uniquely determined):

**Proposition 1.** *Define $R'(\cdot)$ as $\frac{dR}{dd_p^*}$ and $c'(\cdot)$ as $\frac{dc}{dd_p^*}$. For any agent $p$, given Assumptions 1- 4 and an observed $\Delta_p(d_p^*)$,*

$$\lambda_p \in \left[ \frac{R'(\Delta_p(d_p^*))}{c'(\Delta_p(d_p^*))}, \infty \right), \tag{3}$$

*and the bound is sharp.*

Intuitively, different values of the unknown $d_p^*$ yield different estimates of $\lambda_p$ consistent with the observed $\Delta_p(d_p^*)$. Thus, uncertainty in $d_p^*$ results in uncertainty in $\lambda_p$. Equivalently, point-identifying $\lambda_p$ requires *perfect* knowledge of $d_p^*$. Thus, without further assumptions, $\lambda_p$ is only partially identifiable. The lower bound is attained for $d_p^* = 0$ (all $d_i = 1$ are manipulated), while $\lambda_p \to \infty$ as $\Delta_p(d_p^*) \to d_p^*$ (no manipulation). Intuitively, increases in $\lambda_p$ further disincentivize increases to $\Delta_p(d_p^*)$, such that $\Delta_p(d_p^*)$ gets closer to $d_p^*$.

A naïve approach to gaming detection would be to rank individuals using the above bound. To see why this is problematic, consider an Agent 1 ($\lambda_1 = 10$) and Agent 2 ($\lambda_2 = 30$), and let $R(x) = x$ and $c(x) = x^2$. Suppose Agent 1 is a health insurance plan serving a relatively healthy population ($d_1^* = 0.05$), while Agent 2 serves a population with a higher burden of illness ($d_2^* = 0.12$). Via Eq. 2, we have $\Delta_1(d_1^*) = 0.10$, while $\Delta_2(d_2^*) \approx 0.14$. Substitution into Eq. 3 yields $\lambda_1 \geq 5$ and $\lambda_2 \geq 3.66$, flipping the true ranking of $\lambda_p$. Thus, acting on this bound may incorrectly penalize agents when a high $\Delta_p(d_p^*)$ is appropriate; *e.g.*, insurance plans serving sicker populations.

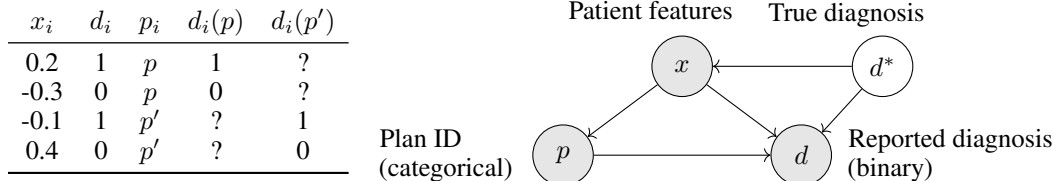

| $x_i$ | $d_i$ | $p_i$ | $d_i(p)$ | $d_i(p')$ |
|-------|-------|-------|----------|-----------|
| 0.2   | 1     | $p$   | 1        | ?         |
| -0.3  | 0     | $p$   | 0        | ?         |
| -0.1  | 1     | $p'$  | ?        | 1         |
| 0.4   | 0     | $p'$  | ?        | 0         |

Figure 2: **Left:** Toy dataset with observed factual outcomes $d_i(p)$ and $d_i(p')$. "?" denotes missing counterfactual outcomes. **Right:** Causal graph for gaming detection with confounders $\mathbf{x}$, agent indicator $p$, ground truth diagnosis $d^*$, and agent decision $d$.

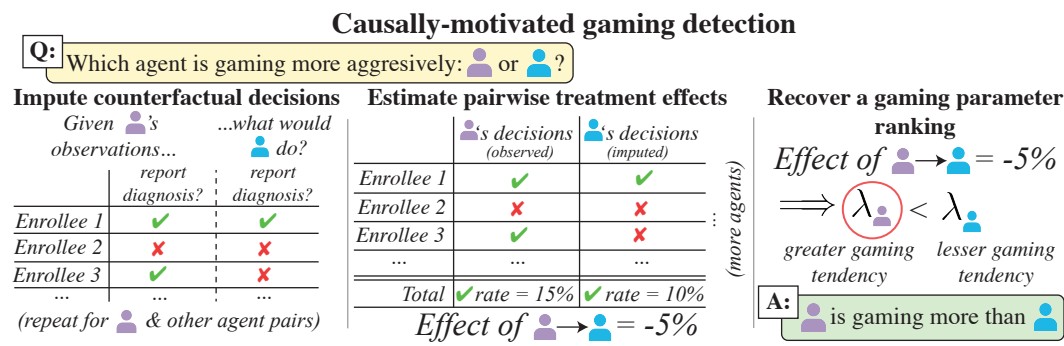

Figure 3: Causally-motivated gaming detection. **Left:** First, we impute counterfactual decisions for each agent. **Middle:** The imputed counterfactuals yield average treatment effects (ATEs) across *pairs* of agents. **Right:** Using ATE estimates to rank agents yields a ranking of the gaming parameter $\lambda_p$. We show one direction of comparison across agents for simplicity. In practice, we impute decisions for both directions of comparison and average (given blue agent's observations, impute purple agent's decisions).

### 4.2 Identifying a ranking of the gaming parameter

Since we showed that point-identifying $\lambda_p$ is impossible without further assumptions, we relax gaming detection to a *ranking* problem. Intuitively, differences in agent behavior under similar conditions may indicate different gaming capacities, from which the proposed approach follows.

**Ranking $\lambda_p$ by estimating counterfactuals.** Recall that we aim to find agents with the lowest $\lambda_p$. Thus, it suffices to *rank* agents by $\lambda_p$, which can be done as follows:

**Theorem 1.** *Under Assumptions 1- 5, and $\Delta_{p'}(d_p^*)$ defined as*

$$\Delta_{p'}(d_p^*) \triangleq \arg\max_{\bar{d} \in [0,1]} R(\bar{d}) - \lambda_{p'} c(\bar{d} - d_p^*), \tag{4}$$

*we have that $\Delta_p(d_p^*) < \Delta_{p'}(d_p^*)$ if and only if $\lambda_p > \lambda_{p'}$.*

Theorem 1 tells us that estimation of $\Delta_p(d_p^*)$ and $\Delta_{p'}(d_p^*)$ can be used to rank $\lambda_p$ vs. $\lambda_{p'}$. Eq. 4 differs from Eq. 2: while $R$, $c$, and $d_p^*$ are the same, $\lambda_p$ changes to $\lambda_{p'}$. While subtle, the distinction is key to gaming detection: $\Delta_{p'}(d_p^*)$ denotes what agent $p'$ *would have* done in a population with ground truth $d_p^*$, as opposed to what agent $p'$ *actually did* in their population (ground truth $d_{p'}^*$).

To build a ranking strategy, first consider ranking two agents $p$ and $p'$. Define $\mathcal{D}_{p,p'} \triangleq \{(\mathbf{x}_i, d_i, p_i) \mid i = 1, \ldots, n \text{ and } p_i \in \{p, p'\}\}$, where $p_i$ is an indicator for the agent that observed example $i$. Following the Neyman-Rubin potential outcomes framework [35], let $d_i(p)$ be the value of $d_i$ if $p_i$ was set to $p$ (*i.e.* had agent $p$ been *compelled* to make a decision). Such variables are called *counterfactuals*. Figure 2 (left) shows a toy dataset with counterfactuals $d_i(p), d_i(p')$, where "?" are unobserved decisions $d_i$. Dropping unobserved data, the average $d_i(p)$ is $\Delta_p(d_p^*)$ by definition. Thus, if $d_i(p)$ and $d_i(p')$ are fully observed, one could estimate $\Delta_p(d_p^*)$ and $\Delta_{p'}(d_p^*)$ as follows:

$$\hat{\Delta}_p(d_p^*) = \frac{1}{n_p} \sum_{\substack{(\mathbf{x}_i, d_i, p_i) \in \mathcal{D}_{p,p'} \\ p_i = p}} d_i(p) \qquad \hat{\Delta}_{p'}(d_p^*) = \frac{1}{n_p} \sum_{\substack{(\mathbf{x}_i, d_i, p_i) \in \mathcal{D}_{p,p'} \\ p_i = p}} d_i(p') \tag{5}$$

| **a)** Causally-motivated gaming detection. | **b)** Python-style pseudocode. |
|---|---|
| **Input:** $\mathcal{D} = \{(\mathbf{x}_i, d_i, p_i)\}_{i=1}^N$ and a list of $P$ agents $\mathbf{p}$ 
 **Output:** sorted list of $P$ agents 
    $\hat{\tau} \leftarrow$ fitted causal effect estimator using $\mathcal{D}$ 
    $\mathbf{T} \leftarrow$ empty $P$-by-$P$ matrix 
    **for all** $p_i, p_j$ where $i < j$ **do** 
      $\mathbf{T}[i,j] \leftarrow \hat{\tau}(p_i, p_j)$ 
    **end for** 
    $\mathbf{\Lambda} \leftarrow$ sort $\mathbf{p}$, using $\mathbf{T}$ to make comparisons 
    **return** $\mathbf{\Lambda}$ | <pre>train_data, test_data = split(dataset)
model = fit_causal_estimator(train_data)
comparisons = {}
for p_i, p_j in agents where p_i != p_j:
    # counterfactual est., plugging in agent IDs
    agent_i_cf = model.predict(test_data, p_i)
    agent_j_cf = model.predict(test_data, p_j)
    comparisons[(p_i, p_j)] = agent_i_cf -
        agent_j_cf

ranking = sort(agents, comparisons) # use the
    comparisons dictionary as a lookup table
return ranking</pre> |

Figure 4: Pseudocode for causally-motivated gaming detection. Causal effect estimators take pairs of agents $(p, p')$ and their observations $\mathbf{x}^{(\cdot)}$ as inputs, and output the mean difference in predicted decision rate.

where $n_p$ is the number of observations by agent $p$, and use these estimates to rank $\lambda_p$ as per Theorem 1. However, only one of $d_i(p)$ or $d_i(p')$ is ever observed. The need to estimate a counterfactual (namely, $\Delta_{p'}(d_p^*)$) suggests a causal inference approach, which proceeds assuming the following [36]:

**Assumption 6** (Conditional exchangeability). *For all $i$, $d_i(p_i) \perp\!\!\!\perp p_i \mid \mathbf{x}_i$, where $d_i(p_i)$ is the potential outcome of $d_i$ under agent $p_i$.*

**Assumption 7** (Consistency). *For all $i$, $d_i(p_i) = d_i$.*

**Assumption 8** (Positivity/overlap). *For any agent $p$ and $\mathbf{x} \in \mathcal{X}$, $0 < \mathbb{P}[p \mid \mathbf{x}] < 1$.*

Via Assumptions 6- 8, we have that $\mathbb{E}[d_i(p) \mid \mathbf{x}_i] = \mathbb{E}[d_i \mid \mathbf{x}_i, p]$ and $\mathbb{E}[d_i(p') \mid \mathbf{x}_i] = \mathbb{E}[d_i \mid \mathbf{x}_i, p']$ [37]. Hence, the causal effect is *identifiable*. This estimand corresponds to a three-variable causal graph, where $\mathbf{x}_i$ are confounders, the agent indicator $p_i$ is a "treatment," and the agent's decision $d_i$ is the outcome (Fig. 2, right).[1] We proceed by estimating the effect of "swapping" agents:

**Corollary 1.** *Define $\tau(p, p') \triangleq \mathbb{E}_{\mathbf{x}_i}[\mathbb{E}[d_i(p) \mid \mathbf{x}_i]] - \mathbb{E}_{\mathbf{x}_i}[\mathbb{E}[d_i(p') \mid \mathbf{x}_i]]$. Then, given Assumptions 1- 8, $\tau(p, p') > 0$ if and only if $\lambda_p < \lambda_{p'}$.*

Since $\mathbb{E}[d_i(p) \mid \mathbf{x}_i] = \mathbb{E}[d_i \mid \mathbf{x}_i, p]$, it is an unbiased estimator of $\Delta_p(d_p^*)$ by definition, and substitution into Theorem 1 yields the result. Thus, one can rank $\lambda_p$ by estimating the effect of switching from agent $p$ to $p'$ on the observed decision rate for *all* pairs of agents $p, p'$. Each effect estimate is a pairwise comparison of agents, from which a ranking of $\lambda_{(\cdot)}$ follows. We summarize causally-motivated gaming detection in Fig. 3, with pseudocode in Fig. 4.

**Obtaining a well-ordered ranking.** Since we aim to rank $\lambda_p$, our chosen estimator should yield a well-ordered ranking. Via Corollary 1, the oracle treatment effect $\tau$ suffices, but $\tau$ must generally be estimated. We show that a "sufficiently" accurate estimate $\hat{\tau}$ also yields the desired result:

**Proposition 2.** *Let $\tau(\cdot)$ be the oracle treatment effect function as defined in Corollary 1, and $\hat{\tau}$ be its sample estimate. Given Assumptions 1- 8, for any $\varepsilon > 0$, if $\sup |\hat{\tau}(p, p') - \tau(p, p')| \leq \varepsilon$, then, for all $p, p'$ such that $\min_{p,p'} |\tau(p, p')| > \varepsilon$, $\hat{\tau}(p, p') > 0$ if and only if $\lambda_p < \lambda_{p'}$.*

The result is immediate: sufficiently low estimation error in $\hat{\tau}$ (*i.e.*, $\leq \varepsilon$) cannot "flip" any pairwise rankings where $\tau > \varepsilon$. Thus, any consistent estimate of $\tau$ yields (asymptotically) a well-ordering of $\lambda_p$. We defer to past asymptotic analyses of causal effect estimation for further discussion [38, 39].

## 5 Empirical results & discussion

We aim to demonstrate that causal inference can be used for gaming detection. First, we discuss our setup (Section 5.1). Then, we show in synthetic data (Section 5.2) that causal methods require fewer audits than existing non-causal methods to catch the worst offenders. Finally, in real-world case study (Section 5.3), we find that causal methods yield rankings correlated with suspected drivers of gaming.

---

[1]Note that, under the causal graph of Figure 2 (right), the non-identifiability of $\lambda_p$ is immediate from the $d$-separation properties of the graph; see Appendix B for further comment.

## 5.1 Setup

We describe the datasets, evaluation method, and gaming detection methods under consideration.

**Datasets.** In real datasets, ground truth gaming rankings are often unavailable. Thus, we validate our framework in a **synthetic dataset**. We hand-select 20 $\lambda_p$ values (one per agent; see Appendix C.1 for raw $\lambda_p$ values) and simulate confounding by generating covariates from agent-specific Gaussians with means $\boldsymbol{\mu}_p \in \mathbb{R}^2$. To control confounding strength, we set $\boldsymbol{\mu}_p = g(\log(\lambda_p))$, where $g$ is an affine transformation such that the range of $\boldsymbol{\mu}_p$ across agents equals a chosen range parameter $R_\mu \in \{0, 0.1, \ldots, 1.0\}$. Smaller $R_\mu$ implies less confounding, since $\boldsymbol{\mu}_{(\cdot)}$ varies less across agents. We generate 500 observations $\mathbf{x}^{(i)} \in \mathbb{R}^2$ and ground-truth $d^{*(i)}$ per agent via

$$\mathbf{x}^{(i)} \sim \mathcal{N}(\boldsymbol{\mu}_{p^{(i)}}, \sigma^2 \mathbf{I}_{2\times 2}) \qquad d^{*(i)} \sim Ber(\alpha^{*(i)}); \quad \alpha^{*(i)} = \sigma(\mathbf{w}^\top \mathbf{x}^{(i)} + b),$$

where $\mathbf{w} \sim \mathcal{U}(0, 1)^2$, such that increasing $\mathbf{x}$ increases $\alpha^{*(i)}$ (and thus $P(d^{(i)} = 1)$), $b$ is chosen such that $\alpha^{*(i)}$ for the mean $\mathbf{x}$ is $\approx 5\%$, and $\sigma^2 = 1$.[2] We simulate gamed agent decisions $d^{(i)}$ as follows:

$$d^{(i)} \sim Ber(\alpha_p^{(i)}); \quad \alpha_p^{(i)} = \underset{\tilde{d}_p \in [0,1]}{\arg\max} \, \log(\tilde{d}_p) - \lambda_p(\tilde{d}_p - d^{*(i)})^2.$$

Recall that the decisions $d^{(i)}$ are also agent inputs to a payout model. We generate 10 datasets (each $N = 10,000$; 20 agents × 500 observations) for all 11 levels of confounding (as measured by the range of means $R_\mu$). Causal inference assumptions hold in the synthetic data: all confounders $\mathbf{x}^{(i)}$ are observed (Assump. 6), consistency holds by construction (Assump. 7), and overlap holds since all $\mathbf{x}^{(i)} \mid p$ are supported on $\mathbb{R}^2$ (Assump. 8). Full synthetic data generation details are in Appendix C.1.

To benchmark causal effect estimation in a more realistic setting, we apply causal methods to gaming detection in **U.S. Medicare** claims. Medicare is the public health insurance system in the U.S. for residents aged 65 and over. Since private insurance claims data is not widely available, we conduct a gaming case study in healthcare providers. In Medicare, the U.S. government pays healthcare providers on a per-service basis [40]. Thus, providers may be incentivized to label enrollees with as many diagnoses as possible to secure extra payment from the government. We select a 0.2% sample of all Medicare enrollees with a claim in 2018; *i.e.*, those who utilized a service covered directly by Medicare ($N = 37,893$). We use demographic information and diagnoses in 2018 as covariates and select the rate of uncomplicated diabetes diagnosis in 2019 as the outcome. Given differences in healthcare policy and access across U.S. states, we pool data at the U.S. state level and treat each state as an "agent." Additional cohort details are in Appendix C.2.

**Evaluating rankings.** Given an observational dataset of the form $\{(\mathbf{x}_i, d_i, p_i)\}_{i=1}^N$, with covariates $\mathbf{x}_i$, observed decisions $d_i$, and agent indicators $p_i$, gaming detection algorithms output an ordinal agent ranking in terms of the gaming parameter $\lambda_p$. We aim to measure the efficiency of a predicted ranking given some level of resources committed by a decision-maker (*i.e.*, # of agents audited).

Ground truth rankings are available in synthetic data. Thus, we measure the top-5 **sensitivity** at $k$ ($S_k$), the % of top-5 worst offenders in the predicted top-$k$, and the **discounted cumulative gain (DCG)** at $k$, a weighted sum of ground-truth "relevance scores" for the top-$k$ predicted agents, across audit intensities $k \in \{1, \ldots, 20\}$. For a $K$-agent dataset, we define the relevance score as $K + 1$ minus the ground-truth rank (*e.g.*, true rank 1 = relevance $K$, true rank 2 = relevance $K - 1$, etc.). Concretely, for $r_i$ defined as the $i$th ranked agent in a predicted ranking, and rank($\cdot$) as the function returning the *ground-truth* ordinal rank with respect to $\lambda_p$, our ranking evaluation metrics are computed as follows:

$$S_k \triangleq \frac{1}{5} \sum_{i=1}^k \mathbb{1}[\text{rank}(r_i) \le 5] \qquad \text{DCG}_k \triangleq \sum_{i=1}^k \frac{K - \text{rank}(r_i)}{\log_2(i+1)}. \tag{6}$$

Note that sensitivity is an all-or-nothing measure of audit quality given a fixed audit intensity $k$. However, DCG rewards higher predicted rankings for top-$k$ worst offenders, regardless of the absolute ranking position. Furthermore, DCG weights decrease with predicted rank (Eq. 6), which prioritizes

---

[2]For ease of implementation, $\mathbf{x}^{(i)}$ is generated conditioned on $p^{(i)}$. This remains consistent with the causal DAG (Fig. 2, right), since the DAG factorizes as $P(D \mid P, X)P(P \mid X)P(X)$, or equivalently, $P(D \mid P, X)P(X \mid P)P(P)$, as in our data-generation process.

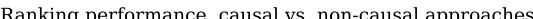

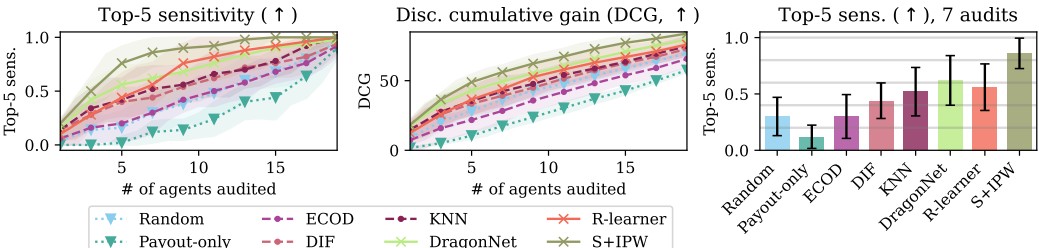

Figure 5: Mean top-5 sensitivity (left) and DCG (center) across # of agents audited, and top-5 sensitivity with 7 audits (right) at mean range 0.9, with $\pm\sigma$ error. Causal methods improve over non-causal baselines. $\nabla$: naïve baselines. $\circ$: anomaly detectors. $\times$: causal effect estimators.

correctly ranking the worst offenders over correctly ranking agents unlikely to game. To summarize audit efficiency across $k$, we also report **area under the top-5 sensitivity curve (AUSC)** across $k$.[3]

In contrast, ground truth gaming rankings are unavailable in the Medicare cohort. As an exploratory analysis, we compare an estimated gaming ranking to 104 state-level healthcare statistics from the 2003-2017 National Neighborhood Data Archive [41] and 2018 Medicare Provider of Service files [42] relating to healthcare access and hospital information (*e.g.*, ownership and size). We report the top five statistics most positively and negatively correlated with our predicted rankings in terms of **Spearman rank-correlation**. State-level summary statistics used are enumerated in Appendix C.2.

**Models.** To demonstrate the utility of causal effect estimation for gaming detection, we compare non-causal baselines to causal effect estimators. Non-causal approaches include a payout-only ranking (based on $P(d_i = 1)$ per agent) and random ranking. We also compare to existing approaches in anomaly detection, which do not make causal assumptions, but assume that gamed decisions are outlier-like. We test $k$-nearest neighbor outlier detection (KNN) [17], empirical-cumulative-distribution-based outlier detection (ECOD) [22], and deep isolation forests (DIF) [23]. These methods use $(\mathbf{x}_i, d_i)$ as inputs to an "anomaly score" model. We use average within-agent anomaly scores as a ranking. We discuss other works in algorithmic anomaly/fraud detection in Appendix A.

We implement the following causal effect estimators. **PSM** fits a propensity score model to match points in one agent's population to its nearest neighbor in the other agent's population with respect to propensity score estimates [43]. The **S-learner** trains one outcome prediction model for all agents, while the **T-learner** trains one model per agent [44]. **DragonNet** jointly models the outcome (one prediction "head" per agent) and propensity score to control for confounding [45]. **S+IPW** fits the S-learner with estimated sample weights that reweight the observed distribution to resemble an unconfounded distribution (randomized treatment assignment) [43]. The **R-learner** fits an outcome and propensity model, then regresses the residuals on one another to obtain a final unconfounded estimator [39]. Hyperparameters for all baselines and causal effect estimators are in Appendix E.

**Implementation details.** We use neural networks for all modeling (causal effect estimators + DIF). We use one-hot encoding for treatments (agent indicators). Matching across pairs of agents occurred without replacement (one-to-one), dropping unmatched individuals. We use generalizations of IPW and R-learners to multiple treatments, namely permutation weighting [46] and the structured intervention network [47], respectively. We perform a 7:3 dataset train-test split, training all models on the larger split. All rankings are computed on the test split. Early stopping is performed on a 20% validation split randomly sampled from the training set. Full modeling and training details, including the architecture and hyperparameters used, are in Appendix E.

## 5.2 Gaming detection in synthetic data

**Causal effect estimators identify gaming more efficiently than non-causal baselines.** Figure 5 shows the top-5 sensitivity and DCG of rankings produced by causal vs. non-causal gaming detection approaches at high confounding (mean range: 0.9). Since the S-, T-learner, and DragonNet perform

---

[3]AUSC is similar to the area under the ROC curve (AUROC) with the top-$k$ worst offenders defined as the "positive class," except that the $x$-axis is the # of audits rather than false positive rate. Unlike AUROC, random performance is less than 0.5, but tends to 0.5 as the number of agents $K \rightarrow \infty$ (Appendix B.6).

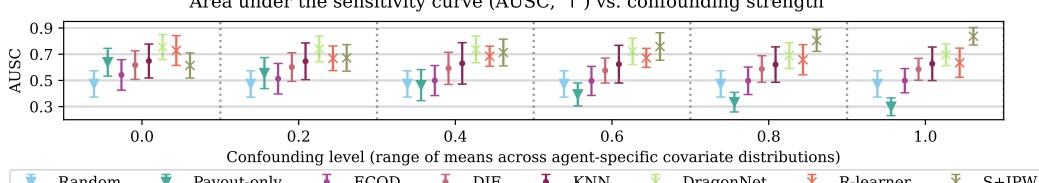

Figure 6: Area under the sensitivity curve (AUSC) for causal vs. non-causal methods across levels of confounding, with $\pm\sigma$ error. As confounding increases, a payout-only ranking degrades. Anomaly detection performance does not vary across confounding strength, maintaining slightly better than random rankings. Causal methods generally maintain higher mean AUSC than baselines across confounding levels. $\triangledown$: naïve baselines. $\circ$: anomaly detectors. $\times$: causal effect estimators.

similarly (Appendix D.2), we show only DragonNet here. Since PSM underperforms due to challenges with multi-treatment confounding control, we also defer results for PSM to Appendix D.2. Across audit intensities, causal approaches outperform non-causal methods in terms of sensitivity (*e.g.*, at 7 audits, Fig. 5, right; S+IPW: 0.860±0.135 vs. KNN: 0.520±0.215) and DCG (S+IPW: 56.1±5.11 vs. KNN: 42.3±10.8).[4] Trends are similar at other levels of confounding (Appendix D.1), though the gain in ranking performance of causal approaches over non-causal methods diminish at lower levels of confounding (Figure 7; S+IPW AUSC: 0.614±0.096 vs. KNN: 0.648±0.129, mean range 0.0).

A payout-only approach yields worse than random ranking with sufficient confounding between covariates and agent decisions (payout-only AUSC: 0.299±0.067 vs. random: 0.473±0.101, Figure 6, mean range 1.0; *e.g.*, if healthier patients are enrolled in more gaming-prone plans). Anomaly detection methods (KNN, ECOD, DIF) are ill-suited for detecting gaming in dense regions of covariate space by design, while causal approaches would excel due to improved overlap. If outliers are more likely to be manipulated (*e.g.*, if populations with lower ground-truth $d_p^*$ are more likely to be gamed), an anomaly detection method would identify gaming in such points. Indeed, anomaly detectors empirically outperform random ranking but lag causal methods. We

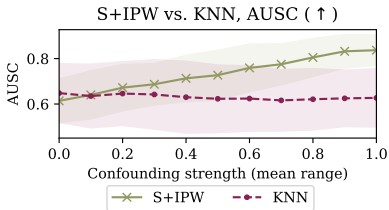

Figure 7: AUSC of S+IPW (causal) vs. KNN (non-causal) across confounding strength with $\pm\sigma$ error. The advantage of S+IPW over KNN decreases as confounding diminishes.

further discuss why anomaly detection methods underperform causal approaches in Appendix D.1.

Trends in ranking performance vary across causal effect estimators. DragonNet and the R-learner degrade slightly as confounding increases (DragonNet AUSC: 0.755±0.096 → 0.696±0.083; R-learner: 0.728±0.114 → 0.635±0.111; mean range 0.0 → 1.0), likely due to slightly worse overlap. However, S+IPW *improves* as confounding increases (AUSC: 0.614±0.096 → 0.837±0.067; mean range 0.0 → 1.0). This is likely since the oracle IPW weights deviate from a uniform weighting as confounding increases. In such settings, estimation error in IPW weights may be less likely to incorrectly up-weight points that an oracle propensity score would down-weight, and vice versa. While such error would bias the pointwise treatment effect estimate, for the purposes of ranking, causal effect estimators only need to correctly estimate the *sign* of the treatment effect. Thus, we hypothesize that our ranking may be less affected by such errors in the propensity score estimate. We leave formal analyses of the properties of IPW with respect to the sign of causal effect estimates to future work. A sensitivity analysis of all causal approaches is in Appendix D.2.

**Takeaways.** In a synthetic dataset, causal effect estimation approaches identified gaming more efficiently than non-causal baselines across levels of confounding. The empirical results provide proof-of-concept for causal effect estimation as a gaming detection method.

### 5.3 Case study: Detecting upcoding in U.S. Medicare

As an exploratory analysis, we use the best-performing approach in the synthetic data (S+IPW) analyze upcoding by U.S. state in U.S. Medicare. Table 1 shows the five state-level healthcare statistics most positively and negatively correlated with gaming rankings predicted by S+IPW.

---

[4]We report $\pm\sigma$ error bars as in the figures.

Table 1: Top 5 features most positively and negatively correlated with state gaming rankings predicted by S+IPW, as measured by Spearman correlation. We report $p$-values given a null hypothesis of zero correlation between predicted rankings and the statistic of interest. OP PT/SP: Outpatient physical therapy and speech pathology. SNF: skilled nursing facility.

| State-level healthcare system statistic | Spearman corr. | $p$-value |
|---|---|---|
| % of OP PT/SP providers that are for-profit | 0.294 | 0.036 |
| ratio of for-profit to non-profit hospitals | 0.272 | 0.053 |
| % of hospice providers that are for-profit | 0.232 | 0.101 |
| % providers that are ambulatory surgery centers | 0.222 | 0.118 |
| % of hospitals that are for-profit | 0.222 | 0.118 |

**Upcoding is correlated with for-profit provider prevalence.** Four of the top five features most positively associated with our predicted ranking reflect a greater state-level prevalence of for-profit healthcare providers. This matches the intuition that for-profit providers may game more aggressively due to stronger profit motives. Notably, the feature 2nd-most positively correlated with our rankings (ratio of for-profit to non-profit hospitals) is a suspected driver of upcoding in Medicare [4, 16], with the idea that competition from for-profit providers drives non-profit providers towards gaming. Note that many of the correlations are not statistically significant, and unmeasured factors such as healthcare quality could explain differences in diagnosis coding, rather than gaming. Despite the limitations, causal effect estimation shows promise as a practical approach to gaming detection.

**Takeaways.** In an exploratory case study of gaming in U.S. Medicare, causal effect estimation yields a ranking of U.S. states that *positively* correlates with the prevalence of for-profit healthcare providers, matching domain expertise on suspected drivers of gaming.

## 6   Conclusion

We propose a causally-motivated framework for ranking agents by gaming propensity in the context of strategic adaptation. We show that the gaming parameter is only partially identifiable, but a ranking of a set of agents based on the gaming deterrence parameter is identifiable via causal inference. We demonstrate the utility of causal effect estimation for gaming detection on synthetic data and a case study of upcoding in Medicare.

**Limitations & broader impact.** We assume agents always increase $d_i$ with respect to ground truth, and that gaming explains all differences in agent behaviors, ignoring factors such as agent "quality" (*e.g.*, quality of care). Utility-maximizing behavior and conditional exchangeability are strong assumptions, but are statistically unverifiable. Many works in game theory and causal inference share these limitations. We caution that policies informed by our framework could reinforce imbalanced power dynamics (*i.e.*, are individual citizens [26] or more powerful entities gaming a model) via extraneous or weaponized accusations of gaming, since not all entities have equal capacity to respond to such claims. In particular, gaming by individuals may potentially reflect structural inequities rather than inherently pathological behavior. To mitigate such risks, we suggest "shadowing" studies (decisions visible, but not acted upon) alongside existing audit mechanisms before adoption.

## Acknowledgements

We thank (in alphabetical order) Amanda Kowalski, Daniel Shenfield, Donna Tjandra, Divya Shanmugan, Dylan Zapzalka, Ezekiel Emanuel, Jung Min Lee, Meera Krishnamoorthy, Sarah Jabbour, and Serafina Kamp for helpful conversations and feedback. Special thanks to Dexiong Chen, Michael Ito, Shengpu Tang, Stephanie Shepard, and Winston Chen for their comments on drafts of this work, to Kathryn Ashbaugh, Michael Shafir, and the Advanced Research Computing team at the University of Michigan for assistance with data access and usage, and Matt Guido for coordinating our meetings. The authors are supported by a grant from Schmidt Futures (Award No. 70960). The funders had no role in the study design, analysis of results, decision to publish, or preparation of the manuscript. This study was deemed exempt and not regulated by the University of Michigan institutional review board (IRBMED; HUM00230364).

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

## A  Additional related works

**Algorithmic anomaly/fraud detection.**   Our framework can be understood as an algorithmic auditing method for fraud/anomaly detection. Many approaches assume that ground-truth fraud/gaming labels are available, reducing gaming detection to supervised learning [48, 49, 50]). We do not assume access to such labels. Unsupervised approaches for anomaly/fraud detection [18, 19, 20, 21] generally assume that anomalies are outliers with respect to some distribution. Mixture-modeling approaches similarly assume that fraudulent/non-fraudulent decisions correspond to distributions learnable under restrictive parametric assumptions [51]. Instead of distributional assumptions, we make behavioral assumptions about agents following strategic classification. Existing models of agent behavior in the context of fraud detection make domain-specific assumptions about features predictive of fraud [52], agent utility (*e.g.*, constant penalties [53]), or access to audit labels [52]. We generalize past work by making looser assumptions on agent utility and does not assume access to auxiliary information (*e.g.*, audit labels), and circumvents the need for ground-truth labels by making assumptions about gaming.

## B  Omitted Proofs

Here, we provide detailed proofs for all theoretical results.

## B.1 Proposition 1

**Proposition.** *Let $R' \triangleq \frac{d}{dd_p^*}$ and $c' \triangleq \frac{d}{dd_p^*}$. For any agent $p$, given Assumptions 1- 4 and a fixed observation of $\Delta_p(d_p^*)$,*

$$\lambda_p \in \left[ \frac{R'(\Delta_p(d_p^*))}{c'(\Delta_p(d_p^*))}, \infty \right). \tag{7}$$

*Proof.* Fix some $R, c$ satisfying assumptions Assumptions 1- 4 and an arbitrary $\Delta_p(d_p^*)$. Note that, for all $\Delta_p(d_p^*) \neq d_p^*$, we can write

$$\lambda_p = \frac{R'(\Delta_p(d_p^*))}{c'(\Delta_p(d_p^*) - d_p^*)}. \tag{8}$$

Note that this quantity is monotonic in $d^*$, since the strict convexity of $c$ implies that $c'$ is strictly increasing. Hence, we can substitute an upper and lower bound on $d^*$ to obtain our desired result. First, since $d^* \in [0, \Delta_p(d_p^*))]$, we can substitute $d^* = 0$ to reach

$$\lambda_p \geq \frac{R'(\Delta_p(d_p^*))}{c'(\Delta_p(d_p^*))}. \tag{9}$$

Before considering the case where $d^* = \Delta_p(d_p^*))$, we note that a direct substitution yields $c'(\Delta_p(d_p^*) - \Delta_p(d_p^*)) = c'(0) = 0$, since $c$ is strictly convex and minimized at 0 via Assumption 3. However, we can use a limiting argument since $c$ is differentiable and therefore continuous:

$$\lim_{d^* \to \Delta_p(d_p^*)^-} \lambda_p = \lim_{d^* \to \Delta_p(d_p^*)^-} \frac{R'(\Delta_p(d_p^*))}{c'(\Delta_p(d_p^*) - d^*)}. \tag{10}$$

To evaluate the limit, we treat $R'$ as a positive constant (via Assumption 2) and note that the limit

$$\lim_{d^* \to \Delta_p(d_p^*)^-} c'(\Delta_p(d_p^*) - d^*) = 0, \tag{11}$$

from which we conclude $\lim_{d^* \to \Delta_p(d_p^*)^-} = \infty$. Combining with the lower bound on $\lambda_p$ yields the desired bounds.

$\square$

**A causal DAG-based perspective on non-identifiability of $\lambda_p$.** In the context of the causal graph of strategic adaptation (Figure 2, right), the non-identifiability of $\lambda_p$ is immediate. Since it is a proxy for the treatment effect of plan ($p$) on reported diagnosis rate ($d$), without accounting for $x$ and $d^*$, conditional exchangeability (Assumption 6) does not hold. The utility-maximization formulation of gaming borrowed from strategic classification provides additional information not encoded in the causal graph. Namely, we have that $R'(\Delta_p(d_p^*))/c'(\Delta_p(d_p^*) - d_p^*)$ is monotonic in $d_p^*$ and $d_p^* \in [0, \Delta_p(d_p^*)]$, provided that $R$ and $c$ satisfy Assumptions 1- 4. These observations allow us to further bound the range of $\lambda_p$.

## B.2 Theorem 1

**Theorem.** *Define $\Delta_{p'}(d_p^*)$ as*

$$\Delta_{p'}(d_p^*) \triangleq \arg\max_{\bar{\mathbf{d}} \in [0,1]} R(\bar{d}) - \lambda_{p'} c(\bar{\mathbf{d}} - d_p^*) \tag{12}$$

*Then, given Assumptions 1- 4, $\Delta_p(d_p^*) < \Delta_{p'}(d_p^*)$ if and only if $\lambda_p > \lambda_{p'}$.*

*Proof.* For all portions of the proof, let $R' \triangleq \frac{d}{dd_p^*}$ and $c \triangleq \frac{d}{dd_p^*}$.

( $\implies$ ) We have the simultaneous first-order conditions

$$R'(\Delta_p(d_p^*)) - \lambda_p c'(\Delta_p(d_p^*) - d^*) = R'(\Delta_{p'}(d_p^*)) - \lambda_{p'} c'(\Delta_{p'}(d_p^*) - d^*) \tag{13}$$

$$\iff \underbrace{R'(\Delta_p(d_p^*)) - R'(\Delta_{p'}(d_p^*))}_{:=C_0 \geq 0} - \lambda_p c'(\Delta_p(d_p^*) - d^*) = -\lambda_{p'} c'(\Delta_{p'}(d_p^*) - d^*) \tag{14}$$

$$\iff \lambda_p c'(\Delta_p(d_p^*) - d^*) - C = \lambda_{p'} c'(\Delta_{p'}(d_p^*) - d^*) \tag{15}$$

$$\iff \lambda_p = \lambda_{p'} \cdot \underbrace{\frac{c'(\Delta_{p'}(d_p^*) - d^*) + C}{c'(\Delta_p(d_p^*) - d^*)}}_{C_1 > 1} > \lambda_{p'}, \tag{16}$$

and hence $\lambda_p > \lambda_{p'}$ as desired. Note that $C_0 \geq 0$ as per Assumption 2, and $C_1 > 1$ as per our assumption that $\Delta_p(d_p^*) < \Delta_{p'}(d_p^*)$, from which we can conclude $c'(\Delta_{p'}(d_p^*) - d^*) > c'(\Delta_p(d_p^*) - d^*)$, because the strict convexity of $c$ implies that $c'$ strictly increases (Assumption 3).

( $\impliedby$ ) Pick any $\lambda_{p'} < \lambda_p$ and fix some $d^*$. By contradiction; suppose $\Delta_p(d_p^*) \geq \Delta_{p'}(d_p^*)$ as well. Then:

$$R'(\Delta_p(d_p^*)) - \lambda_p c'(\Delta_p(d_p^*) - d^*) = R'(\Delta_{p'}(d_p^*)) - \lambda_{p'} c'(\Delta_{p'}(d_p^*) - d^*) \tag{17}$$

$$\iff R'(\Delta_p(d_p^*)) - R'(\Delta_{p'}(d_p^*)) = \lambda_p c'(\Delta_p(d_p^*) - d^*) - \lambda_{p'} c'(\Delta_{p'}(d_p^*) - d^*). \tag{18}$$

Now, note that $R'(\Delta_p(d_p^*)) - R'(\Delta_{p'}(d_p^*)) \leq 0$, since $\Delta_p(d_p^*) \geq \Delta_{p'}(d_p^*)$ by assumption and $R'$ is non-increasing in its argument. Furthermore, we can write

$$\lambda_p c'(\Delta_p(d_p^*) - d^*) - \lambda_{p'} c'(\Delta_{p'}(d_p^*) - d^*) > \lambda_{p'}(c'(\Delta_p(d_p^*) - d^*) - c'(\Delta_{p'}(d_p^*) - d^*)) > 0, \tag{19}$$

where the first inequality is due to $\lambda_{p'} < \lambda_p$ and factoring, and the second inequality is since $c'(\Delta_p(d_p^*) - d^*) - c'(\Delta_{p'}(d_p^*) - d^*) > 0$ (via Assumption 3) and $\lambda_{(\cdot)} > 0$ (by definition). Returning to Eq. 18, we can write

$$0 \geq R'(\Delta_p(d_p^*)) - R'(\Delta_{p'}(d_p^*)) = \lambda_{p'} c'(\Delta_{p'}(d_p^*) - d^*) > 0 \tag{20}$$

which is a contradiction ($0 \not> 0$). Thus, $\lambda_{p'} < \lambda_p$ implies $\Delta_p(d_p^*) < \Delta_{p'}(d_p^*)$ as desired. $\qquad\square$

**Potential extensions to upcoding detection.**     With stronger assumptions, a weaker form of upcoding detection, which is stronger than ranking, is possible:

**Assumption 9** (Known cost and reward derivatives).  *$R'$ and $c'$ can be evaluated at arbitrary points.*

**Assumption 10** (Uncertainty in $d_p^*$ is bounded).  *Lower and upper bounds on $d_p^*$ are possible to obtain.*

Then, define the following:

**Definition 1** ($\varepsilon$-gaming).  *For $\varepsilon > 0$, an agent $p$ is $\varepsilon$-gaming if $\Delta_p(d_p^*) - d_p^* > \varepsilon$.*

In other words, we can define a threshold based on some $\varepsilon$ tolerance in deviations from $d_p^*$ for detecting gaming. Now, recall that

$$\lambda_p = \frac{R'(\Delta_p(d_p^*))}{c'(\Delta_p(d_p^*) - d_p^*)}, \tag{21}$$

which is *non-increasing* in $\Delta_p(d_p^*)$ due to the concavity of $R$ and strict convexity of $c$. Thus, substituting, we could define an agent-specific threshold

$$\lambda^*(p) = \frac{R'(\varepsilon + d_p^*)}{c'(\varepsilon)}, \tag{22}$$

and if some procedure for estimating $\lambda_p$ yields an estimate $\hat{\lambda}_p < \lambda^*(p)$, we conclude that the agent is $\varepsilon$-gaming, while $\hat{\lambda}_p > \lambda^*(p)$ rules out gaming.

$$\lambda^*(p) = \frac{R'(\varepsilon + d_p^*)}{c'(\varepsilon)}, \tag{23}$$

However, $d_p^*$ is still unknown. Suppose that, we can bound $d_p^* \in [\underline{d}_p, \overline{d}_p]$ with probability $1 - \delta$: (*e.g.*, via a parametric binomial proportion confidence interval about some estimate of $d_p^*$):

$$\lambda^*(p) \in \left[ \frac{R'(\varepsilon + \overline{d_p^*})}{c'(\varepsilon)}, \frac{R'(\varepsilon + \underline{d_p^*})}{c'(\varepsilon)} \right], \tag{24}$$

and abbreviate these bounds to $\lambda^*(p) \in [\underline{\lambda}^*(p; \varepsilon), \overline{\lambda}^*(p; \varepsilon)]$. We can use the same bounds on $d_p^*$ to produce a range of estimates for $\lambda_p$:

$$\hat{\lambda}_p \in \left[ \frac{R'(\Delta_p(d_p^*))}{c'(\Delta_p(d_p^*) - \overline{d}_p^*)}, \frac{R'(\Delta_p(d_p^*))}{c'(\Delta_p(d_p^*) - \underline{d}_p^*)} \right], \tag{25}$$

and abbreviate these bounds to $\hat{\lambda}_p \in [\underline{\hat{\lambda}}_p, \overline{\hat{\lambda}}_p]$. Then, we can compare the intervals for $\lambda^*(p; \varepsilon)$ and $\hat{\lambda}_p$ as follows:

- If $\overline{\hat{\lambda}}_p < \underline{\lambda}^*(p; \varepsilon)$, then agent $p$ is $\varepsilon$-gaming with probability $1 - \delta$.

- If $\underline{\hat{\lambda}}_p > \overline{\lambda}^*(p; \varepsilon)$, then we can rule out that agent $p$ is $\varepsilon$-gaming with probability $1 - \delta$.

- Otherwise, $\hat{\lambda}_p \in [\underline{\hat{\lambda}}_p, \overline{\hat{\lambda}}_p] \cap [\underline{\lambda}^*(p; \varepsilon), \overline{\lambda}^*(p; \varepsilon)]$ is non-empty, and we cannot definitively rule out or prove the existence of $\varepsilon$-gaming.

However, this algorithm may be of purely technical interest: it is doubtful that the requisite assumptions are satisfied in our motivating setting (upcoding detection), especially Assumption 9, and it is similarly unclear whether such assumptions apply in other settings. In addition, if agents game similarly (*i.e.*, have similar values of $\lambda_p$), it may be difficult to rule out/prove $\varepsilon$-gaming for the vast majority of agents.

## B.3   Corollary 1

**Corollary.** *Define $\tau(p, p')$ as above. Then, given Assumptions 1- 8, $\tau(p, p') > 0$ if and only if $\lambda_p < \lambda_{p'}$.*

*Proof.* It is sufficient to show that $\tau(p, p') > 0$ if and only if $\Delta_p(d_p^*) < \Delta_{p'}(d_p^*)$, from which Theorem 1 yields the desired result. We can do so by showing (without loss of generality) that $\mathbb{E}[\mathbb{E}[d_i \mid p, x_i]]$ is an unbiased estimate of $\Delta_p(d_p^*)$ (and the case for $\Delta_{p'}(d_p^*)$ proceeds symmetrically). Since $\Delta_p(d_p^*)$ is equivalent to $\mathbb{P}[d_i = 1 \mid p]$ (Eq. 2), the result is immediate:

$$\mathbb{E}[\mathbb{E}[d_i \mid p, x_i]] = \mathbb{E}[d_i \mid p] = \mathbb{P}[d_i = 1 \mid p] = \Delta_p(d_p^*). \tag{26}$$

$\square$

## B.4   Proposition 2

**Proposition.** *Let $\tau(\cdot)$ be the oracle treatment effect function, and $\hat{\tau}$ be some sample estimate of $\tau$. Given Assumptions 1- 8, for any $\varepsilon > 0$, if $\sup |\hat{\tau}(p, p') - \tau(p, p')| \le \varepsilon$, then, for all $p, p'$ such that $\inf_{p, p'} |\tau(p, p')| > \varepsilon$, $\hat{\tau}(p, p) > 0$ if and only if $\lambda_p < \lambda_{p'}$.*

*Proof.* Choose $\tau, \hat{\tau}$, and some arbitrary $\varepsilon$ as specified in the theorem statement. It suffices to show that for all $p, p'$ such that $\inf_{p, p'} |\tau(p, p')| > \varepsilon$, it holds that $\hat{\tau}(p, p') > 0$ if and only if $\tau(p, p') > 0$.

( $\implies$ ) By contradiction; suppose that $\hat{\tau}(p, p') > 0$ but $\tau(p, p') < 0$. Then, by assumption, $\tau(p, p') < -\varepsilon$. But $\sup |\hat{\tau}(p, p') - \tau(p, p')| \le \varepsilon$, so $\hat{\tau}(p, p') < 0$, yielding a contradiction. Thus, $\hat{\tau}(p, p') > 0 \implies \tau(p, p') > 0$. The reverse direction ( $\impliedby$ ) proceeds identically. Thus, the proposition is true. $\square$

**Connections to robustness to non-rational actors.** We assume throughout that agents behave rationally; *i.e.*, always perfectly maximize the utility function given by Eq. 2. However, the rational actor assumption may not always hold; *e.g.*, if agents do not have the resources to carry out the resource maximizing action, or incorrectly estimate their costs. The former is particularly salient for our motivating problem of Medicare upcoding, in which representatives of certain plans may shedule a home visit with a healthcare professional to generate diagnosis codes [54]—a potentially resource-intensive process. Our ranking formulation can afford some robustness to violations of the rational actor assumption:

**Remark 2** (Robustness to bounded rationality violations). *Let $\Delta_p(d_p^*)$ be the utility-maximizing action, and let $\tilde{\Delta}_p(d_p^*)$ be an agent's observed action, such that $|\Delta_p(d_p^*) - \tilde{\Delta}_p(d_p^*)| < \varepsilon_1/2$ for some $\varepsilon_1 > 0$. Let $\tilde{\tau}(p, p)$ be some sample estimate of $\tau$, fitted via $\tilde{\Delta}(\cdot)$. Then:*

$$|\hat{\tau}(p, p') - \tilde{\tau}(p, p')| = |(\Delta_p(d_p^*) - \tilde{\Delta}_p(d_p^*)) - (\Delta_{p'}(d_p^*) - \tilde{\Delta}_{p'}(d_p^*))| \leq \varepsilon_1. \tag{27}$$

*Then, if* $\sup |\hat{\tau}(p, p') - \tau(p, p')| \leq \varepsilon_2$ *such that* $\varepsilon \triangleq \varepsilon_1 + \varepsilon_2$, *we can apply Proposition 2 directly to conclude that* $\hat{\tau}(p, p') > 0$ *if and only if* $\lambda_p < \lambda_{p'}$.

Intuitively, violations of the rational actor assumption are another source of noise in the estimation of $\tau$. If the noise due to rationality violations plus noise due to standard estimation error are low, then no rankings are flipped, as desired.

## B.5 Identifiability result

For completeness, we show the derivation of the standard causal effect identifiability result for our problem setting (*i.e.*, as in [37]). We note that this is a direct application of a known result in the literature.

**Proposition.** *Given Assumptions 6- 8,* $\mathbb{E}[d_i(p) \mid x_i] = \mathbb{E}[d_i \mid x_i, p]$ *and* $\mathbb{E}[d_i(p') \mid x_i] = \mathbb{E}[d_i \mid x_i, p']$.

*Proof.* We show that $\mathbb{E}[d_i(p) \mid x_i] = \mathbb{E}[d_i \mid x_i, p]$; the case for $\mathbb{E}[d_i(p') \mid x_i]$ proceeds identically. We can write

$$\mathbb{E}[d_i(p) \mid x_i] = \mathbb{E}[d_i(p) \mid x_i, p] = \mathbb{E}[d_i \mid x_i, p] \tag{28}$$

where the first equality is an application of conditional exchangeability (Assumption 6), and the second is an application of consistency (Assumption 7). Positivity (Assumption 8) ensures that the conditional expectations are well-defined. □

## B.6 What is the expected AUSC?

For completeness, we also analyze the AUSC metric and provide a closed-form expression for its expected value under a random ranking. First, consider a population of $K \in \mathbb{N}$ agents, where we are interested in top-$k$ sensitivity for some $k \in \{1, \ldots, K\}$. Suppose that we conduct $m$ random audits, for some $m \in \{1, \ldots, K\}$.

The number of ground truth top-$k$ agents that are in the set of $m$ audited agents can be modeled as a hypergeometric random variable $n_m \sim Hyp(k, K, m)$, which is a variable describing the number of "successes" observed by a random draw from a population of $K$ objects with $k$ "success states" in $m$ draws *without replacement*. By standard properties of the hypergeometric distribution, we know that $\mathbb{E}[n_k] = km/K$. Thus, by linearity of expectation, the average (across audit intensities $m$) number of agents identified by random auditing is given by

$$\frac{1}{K} \sum_{m=1}^{K} \mathbb{E}[n_m] = \frac{k}{K^2} \cdot \frac{K(K-1)}{2} = \frac{k(K-1)}{2K}. \tag{29}$$

We divide the result by $k$ to obtain a *proportion* of the top-$k$ identified (as used in the definition of top-$k$ sensitivity), which yields

$$\frac{1}{2} \cdot \frac{K-1}{K}. \tag{30}$$

For a finite number of agents, this is bounded above by 0.5, but approaches 0.5 as $K \to \infty$ (a population of infinite agents).

## C  Data Processing

### C.1  Fully synthetic data

**Overview.**  Our general fully synthetic data-generation pipeline is as follows. First, for each agent, we draw some value of $\boldsymbol{\mu}_p \in \mathbb{R}^2$. We draw individual observations $\mathbf{x}^{(i)} \in \mathbb{R}^2$ for each agent from a Gaussian with mean $\boldsymbol{\mu}_p$ and some fixed variance $\sigma^2$. Given the $\mathbf{x}^{(i)}$, we simulate a "ground-truth" decision $d^{*(i)}$. Note that $d^{*(i)}$ represents the ground truth decision rate. Then, for each agent $p$, we simulate a "gamed" (*i.e.*, strategically perturbed) version of $d^{(i)}$ via a version of the agent utility function in Eq. 2. We now formally describe the data-generation process.

**Generating a "population" for each agent.**  To generate a "population" upon which each agent makes decisions, we randomly draw a mean vector specifying a Gaussian distribution for each agent. Given $P$ agents, the mean depends on $\lambda_p$ as follows:

$$\boldsymbol{\mu}_p = R_\mu \cdot \left( \frac{\tilde{\lambda}_p - \min_{p'} \tilde{\lambda}_p}{\max_{p'} \tilde{\lambda}_p - \min_{p'} \tilde{\lambda}_p} \right) + b; \quad \tilde{\lambda}_p = \log(\lambda_p), \ \bar{\lambda} = \frac{1}{P} \sum_{i=1}^{P} \tilde{\lambda}_p, \qquad (31)$$

where $a > 0$ is a parameter controlling the range of agent-specific means, and $b \in \mathbb{R}$ is some constnat offset. In other words, we log-transform the $\lambda_p$ values, then apply min-max scaling and a constant shift. This design allows us to control the level of confounding in the synthetic data by changing $R_\mu$. We use $\boldsymbol{\mu}_p$ to generate populations for each agent as described below.

Concretely, we consider $\lambda_{(\cdot)} \in [0.001, 0.003, 0.005, 0.007, 0.009, 0.01, 0.015, 0.02, 0.025, 0.03,$ $0.035, 0.04, 0.045, 0.05, 0.06, 0.07, 0.08, 0.09, 0.1, 0.2, 0.3]$ (20 agents), and set $b = -1$. In practice, this means that $\boldsymbol{\mu}_p \in [-1, R_\mu - 1]$. We manually chose $\lambda_{(\cdot)}$ values to set the difficulty of ranking agents by such that a payout-only approach would succeed under low confounding, but fail under high levels of confounding.

**Generating ground-truth and gamed decisions.**  The covariates $\mathbf{x}^{(i)}$ and ground-truth decisions $d^{*(i)}$ are drawn as per

$$\mathbf{x}^{(i)} \sim \mathcal{N}(\boldsymbol{\mu}_{p^{(i)}}, \sigma^2 \mathbf{I}_{2 \times 2})$$
$$d^{*(i)} \sim Ber(\alpha^{*(i)}); \quad \alpha^{*(i)} = \sigma(\mathbf{w}^\top \mathbf{x}^{(i)} + b_d),$$

where $\mu_{(\cdot)}$ is generated for each agent $p^{(i)}$ as described previously, $\mathbf{w}$ is a randomly generated positive vector, and $b_d \in \mathbb{R}$ is some offset.

The $p^{(i)}$s are assigned deterministically; *i.e.*, we sequentially draw a fixed number of $\mathbf{x}^{(i)}$ from each agent-specific distribution and concatenate the results. For realism, inspired by the health insurance setting, and the fact that diagnosis rates for most conditions are relatively low, we set $b_d$ to $\text{logit}(0.05) - \hat{\mathbb{E}}[\mathbf{w}^\top \mathbf{x}]$ such that $\alpha^{*(i)}$ is relatively low, where $\hat{\mathbb{E}}[\cdot]$ denotes the sample mean.

Lastly, we simulate gamed agent decisions $d^{(i)}$ by solving a per-agent utility maximization problem:

$$d^{(i)} \sim Ber(\alpha_p^{(i)}); \quad \alpha_p^{(i)} = \underset{\tilde{d}_p \in [0,1]}{\arg\max} \ \log(\tilde{d}_p) - \lambda_p(\tilde{d}_p - d^{*(i)})^2.$$

**Implementation details.**  All randomness is seeded *once* at the start of the entire data-generation process.

### C.2  Medicare cohort

We select a 0.2% pseudo-random sample of all U.S. Medicare beneficiaries using the last characters of encrypted beneficiary IDs for inclusion in the cohort. As covariates, we choose age, racial category, biological sex, and diagnosis code categories (defined using the 2018 version of the "Hierarchical Condition Category" [HCC] schema released by the Center for Medicare Services). This yields a total of 97 features.

We exclude enrollees who did not generate any claims (no healthcare utilization; *i.e.*, zero cost as recorded by Medicare), those not located in the 50 U.S. states and the District of Columbia, as well

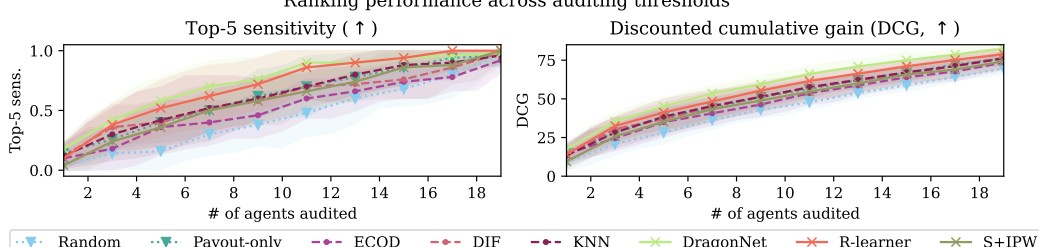

Figure 8: Mean top-5 sensitivity (left) and DCG (right) across # of agents audited at mean range 0.0, with $\pm\sigma$ error. $\triangledown$: naïve baseline. $\circ$: anomaly detection method. $\times$: causal effect estimator.

as *dual-eligible* beneficiaries. Dual-eligibility refers to individuals simultaneously eligible for U.S. Medicare and U.S. Medicaid. While eligibility for Medicare is primarily based on age, eligibility for Medicaid is primarily based on disability status. Dual-eligible beneficiaries are excluded since the U.S. government uses a different payout model for dual-eligible vs. non-dual-eligible enrollees, potentially violating Assumption 1 (shared rewards), since agents may not be reacting to the same payout model for all enrollees.

**Licensing.** Our cohort is drawn from a 20% sample of all U.S. Medicare beneficiaries provided to the authors under a data usage agreement with the Center for Medicare & Medicaid Services.

**State-level healthcare statistics.** We use a mix of raw and engineered features from the CMS (Center for Medicare & Medicaid Services) Provider of Service file (license: Public Use File) [42][5] and the National Neighborhood Data Archive (NANDA; CC-BY 4.0) [41]. In NANDA, data is already aggregated at the state level (including the District of Columbia). We keep statistics pertaining to per-capita or per-sq. mi. healthcare provider density (50 features), filtering to providers with non-zero sales. From the CMS Provider of Service file, data is reported at the provider level. First, we filter out providers ineligible for Medicare participation, and providers that are no longer active. We then manually code ownership information following the provided data dictionaries (for-profit vs. non-profit vs. publicly owned). Next, we compute at the state level the prevalence of for-profit, non-profit, and publicly-owned providers of each type (as defined by CMS) by state, as well as the ratio of for-profit to non-profit providers. Additionally, we extract the average per-hospital bed count, physician count, medical school affiliation rate, and compliance rate as certified by CMS, for a total of 54 features. All aggregations for the Provider of Service file are unweighted (*i.e.*, all hospitals/other healthcare facilities contribute equally). This yields a total of 104 state-level health indicators.

We report the statistics derived from the provider of service file (Table 2) and NANDA (Table 3), with summary statistics across states. Summary statistics are computed excluding infinite and NaN values (*i.e.*, ratios of 0/0 or 1/0). When computing correlations, we exclude states with NaN values for that feature (*i.e.*, ratio of 0/0). Proportions of publicly-owned, non-profit, and for-profit providers do not sum to 1, because providers reporting "unknown" or "other" ownership are excluded. For further information on the feature definitions, consult the data dictionaries for the provider of service[6] and NANDA[7] files.

## D  Supplementary results

### D.1  Causal vs. non-causal approaches, all audit thresholds & all levels of confounding

We show plots with the top-5 sensitivity and DCG at all levels of confounding evaluated, as measured by the mean range $R_\mu$ (*i.e.*, $R_\mu \triangleq \max \mu_p - \min \mu_p$ across agents). We choose $R_\mu \in \{0, 0.1, \dots, 1.0\}$. For convenience, we provide an index of results:

---

[5]Full information on Public Use File licensing as defined by CMS is here: https://www.cms.gov/research-statistics-data-and-systems/files-for-order/nonidentifiabledatafiles

[6]https://www.nber.org/research/data/provider-services-files

[7]https://www.openicpsr.org/openicpsr/project/120907/version/V3/view

| Feature name | Mean±s.d. |
|---|---|
| bed count | 95.125±27.826 |
| % affiliated with a medical school | 3.385± 0.350 |
| # of personnel | 87.182±47.477 |
| % non-compliant | 0.115±0.074 |
| % of providers that are non-profit hospitals | 0.055±0.030 |
| % of providers that are for-profit hospitals | 0.024± 0.015 |
| % of providers that are publicly-owned hospitals | 0.028±0.027 |
| % of providers that are non-profit skilled nursing facilities | 0.069±0.046 |
| % of providers that are for-profit skilled nursing facilities | 0.171±0.068 |
| % of providers that are publicly-owned skilled nursing facilities | 0.020±0.021 |
| % of providers that are non-profit home health agencies | 0.033±0.018 |
| % of providers that are for-profit home health agencies | 0.104±0.075 |
| % of providers that are publicly-owned home health agencies | 0.008±0.010 |
| % of providers that are non-profit OP PT/SP providers | 0.005±0.010 |
| % of providers that are for-profit OP PT/SP providers | 0.025±0.018 |
| % of providers that are publicly-owned OP PT/SP providers | 0.001±0.001 |
| % of providers that are non-profit end-stage renal disease facilities | 0.014±0.012 |
| % of providers that are for-profit end-stage renal disease facilities | 0.096±0.042 |
| % of providers that are publicly-owned end-stage renal disease facilities | 0.001±0.002 |
| % of providers that are non-profit ICF/IDs | 0.041±0.052 |
| % of providers that are for-profit ICF/IDs | 0.025±0.050 |
| % of providers that are publicly-owned ICF/IDs | 0.005±0.007 |
| % of providers that are non-profit rural health clinics | 0.044±0.044 |
| % of providers that are for-profit rural health clinics | 0.024±0.028 |
| % of providers that are publicly-owned rural health clinics | 0.014±0.020 |
| % of providers that are non-profit ambulatory surgery centers | 0.004±0.005 |
| % of providers that are for-profit ambulatory surgery centers | 0.084±0.061 |
| % of providers that are publicly-owned ambulatory surgery centers | 0.002±0.004 |
| % of providers that are non-profit hospice care facilities | 0.020±0.012 |
| % of providers that are for-profit hospice care facilities | 0.039±0.033 |
| % of providers that are publicly-owned hospice care facilities | 0.002±0.004 |
| % of providers that are non-profit | 0.285±0.139 |
| % of providers that are for-profit | 0.592±0.154 |
| % of providers that are publicly-owned | 0.081±0.066 |
| % of urbanized population | 0.679±0.228 |
| % of providers that are ambulatory surgery centers | 0.091±0.064 |
| % of providers that are end-stage renal disease facilities | 0.111±0.040 |
| % of providers that are home health agencies | 0.145±0.069 |
| % of providers that are hospitals | 0.118±0.038 |
| % of providers that are hospice care facilities | 0.070±0.035 |
| % of providers that are ICF/IDs | 0.073±0.088 |
| % of providers that are OP PT/SP providers | 0.007±0.009 |
| % of providers that are rural health clinics | 0.082±0.067 |
| % of providers that are skilled nursing facilities | 0.261±0.073 |
| ratio of for-profit to non-profit providers | 2.838±2.024 |
| ratio of for-profit to non-profit hospitals | 0.730±0.955 |
| ratio of for-profit to non-profit skilled nursing facilities | 3.633±2.387 |
| ratio of for-profit to non-profit home health agencies | 5.135±6.020 |
| ratio of for-profit to non-profit OP PT/SP providers | 10.149*±10.108 |
| ratio of for-profit to non-profit end-stage renal disease facilities | 15.905*±18.546 |
| ratio of for-profit to non-profit ICF/IDs | 1.074*±1.969 |
| ratio of for-profit to non-profit rural health clinics | 0.936±1.380 |
| ratio of for-profit to non-profit ambulatory surgery centers | 32.826*±30.949 |
| ratio of for-profit to non-profit hospice care facilities | 3.989±6.797 |

Table 2: Features chosen for analysis derived from the provider of service file, with mean and standard deviation across states. S.d.: standard deviation. OP PT/SP: outpatient physical therapy and speech pathology. ICF/ID: intermediate care facility for individuals with intellectual disability. *: invalid (1/0 or 0/0) ratios dropped.

| Feature name | Mean±s.d. |
|---|---|
| ambulatory health care specialists per 1000 people | 4.429±2.745 |
| ambulatory health care specialists per sq. mi. | 15.480±16.928 |
| physicians per 1000 people | 1.753±1.196 |
| physicians per sq. mi. | 6.477±7.593 |
| physicians (excluding mental health) per 1000 people | 1.656±1.090 |
| physicians (excluding mental health) per sq. mi. | 6.137±7.195 |
| mental health physicians per 1000 | 0.096±0.112 |
| mental health physicians per sq. mi. | 0.340±0.412 |
| dentists per 1000 people | 0.639±0.242 |
| dentists per sq. mi. | 2.553±2.945 |
| all other health practitioners (besides the above) per 1000 people | 1.483±1.039 |
| all other health practitioners (besides the above) per sq. mi. | 4.949±5.264 |
| chiropractors per 1000 people | 0.282±0.257 |
| chiropractors per sq. mi. | 0.840±0.999 |
| optometrists per 1000 people | 0.100±0.026 |
| optometrists per sq. mi. | 0.307±0.240 |
| non-physician mental health practitioners per 1000 | 0.176±0.390 |
| non-physician mental health practitioners per square | 0.687±0.929 |
| P/O/ST and audiologists per 1000 people | 0.133±0.054 |
| P/O/ST and audiologists per sq. mi. | 0.474±0.657 |
| other health practitioners (besides the above) per 1000 people | 0.792±0.610 |
| other health practitioners (besides the above) per sq. mi. | 2.640±2.742 |
| outpatient care centers per 1000 people | 0.181±0.133 |
| outpatient care centers per sq. mi. | 0.556±0.516 |
| diagnostic labs per 1000 people | 0.075±0.114 |
| diagnostic labs per sq. mi. | 0.229±0.230 |
| home health services per 1000 people | 0.173±0.112 |
| home health services per sq. mi. | 0.517±0.441 |
| other ambulatory care services (besides the above) per 1000 people | 0.125±0.327 |
| other ambulatory care services (besides the above) per sq. mi. | 0.198±0.176 |
| all nursing and residential care facilities per 1000 people | 0.334±0.324 |
| all nursing and residential care facilities per sq. mi. | 0.836±0.650 |
| nursing care facilities per 1000 people | 0.224±0.330 |
| nursing care facilities per sq. mi. | 0.473±0.308 |
| residential IDD care facilities per 1000 people | 0.026±0.014 |
| residential IDD care facilities per sq. mi. | 0.102±0.174 |
| inpatient facilities for care of individuals with IDDs per 1000 people | 0.012±0.010 |
| inpatient facilities for care of individuals with IDDs per sq. mi. | 0.039±0.068 |
| inpatient facilities providing MHSA care per 1000 people | 0.014±0.007 |
| inpatient facilities providing MHSA care per sq. mi. | 0.063±0.109 |
| continuing care and assisted living facilities per 1000 people | 0.023±0.011 |
| continuing care and assisted living per sq. mi. | 0.070±0.062 |
| other residential care facilities (besides the above) per 1000 people | 0.061±0.031 |
| other residential care facilities (besides the above) per sq. mi. | 0.191±0.179 |
| pharmacies and drug stores per 1000 people | 0.314±0.669 |
| pharmacies and drug stores per sq. mi. | 0.853±1.336 |
| optical goods stores per 1000 people | 0.076±0.025 |
| optical goods stores per sq. mi. | 0.282±0.360 |
| misc. other health and personal care stores per 1000 people | 0.054±0.014 |
| misc. other health and personal care stores per sq. mi. | 0.149±0.090 |

Table 3: Features chosen for analysis derived from NANDA, with mean and standard deviation across states. All chosen summary statistics are for providers with > $0 U.S. dollars in sales. S.d.: standard deviation. Sq. mi.: square mile. P/O/ST: physical, occupational, and speech therapists. IDD: intellectual and developmental disabilities. MHSA: mental health and substance abuse.

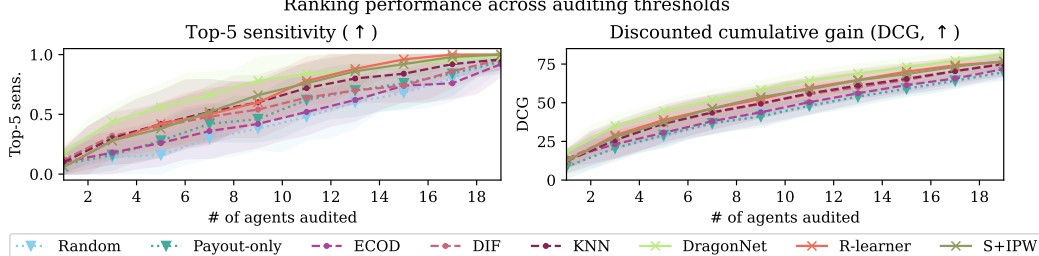

Figure 9: Mean top-5 sensitivity (left) and DCG (right) across # of agents audited at mean range 0.1, with $\pm\sigma$ error. $\triangledown$: naïve baseline. $\circ$: anomaly detection method. $\times$: causal effect estimator.

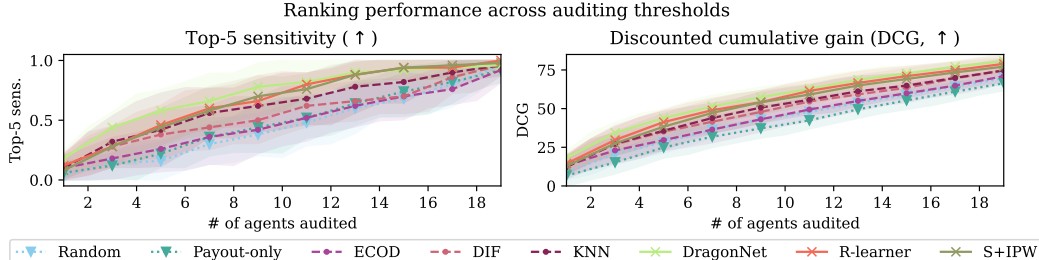

Figure 10: Mean top-5 sensitivity (left) and DCG (right) across # of agents audited at mean range 0.2, with $\pm\sigma$ error. $\triangledown$: naïve baseline. $\circ$: anomaly detection method. $\times$: causal effect estimator.

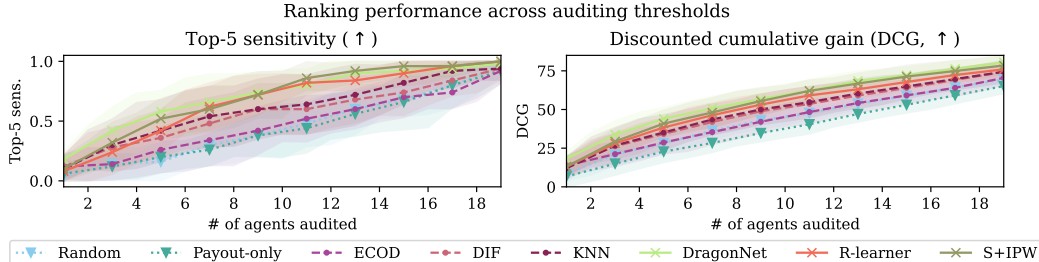

Figure 11: Mean top-5 sensitivity (left) and DCG (right) across # of agents audited at mean range 0.3, with $\pm\sigma$ error. $\triangledown$: naïve baseline. $\circ$: anomaly detection method. $\times$: causal effect estimator.

Figure 12: Mean top-5 sensitivity (left) and DCG (right) across # of agents audited at mean range 0.4, with $\pm\sigma$ error. $\triangledown$: naïve baseline. $\circ$: anomaly detection method. $\times$: causal effect estimator.

- $R_\mu = 0.0$: Figure 8
- $R_\mu = 0.1$: Figure 9
- $R_\mu = 0.2$: Figure 10
- $R_\mu = 0.3$: Figure 11
- $R_\mu = 0.4$: Figure 12
- $R_\mu = 0.5$: Figure 13

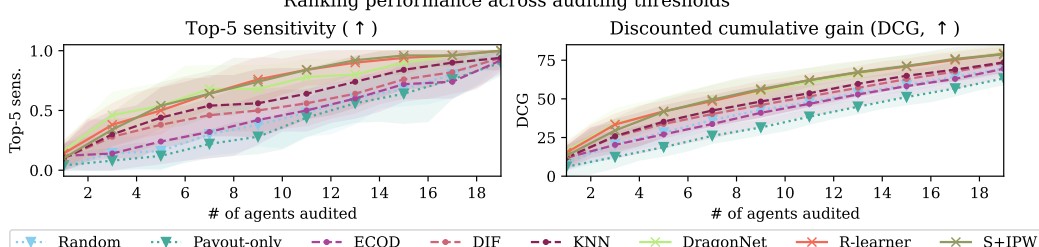

Figure 13: Mean top-5 sensitivity (left) and DCG (right) across # of agents audited at mean range 0.5, with $\pm\sigma$ error. $\nabla$: naïve baseline. $\circ$: anomaly detection method. $\times$: causal effect estimator.

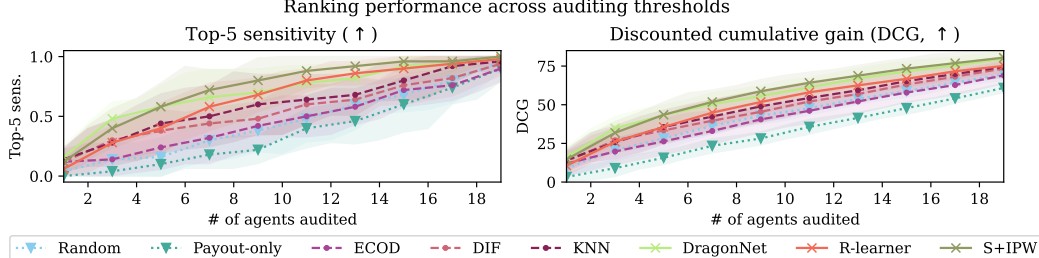

Figure 14: Mean top-5 sensitivity (left) and DCG (right) across # of agents audited at mean range 0.6, with $\pm\sigma$ error. $\nabla$: naïve baseline. $\circ$: anomaly detection method. $\times$: causal effect estimator.

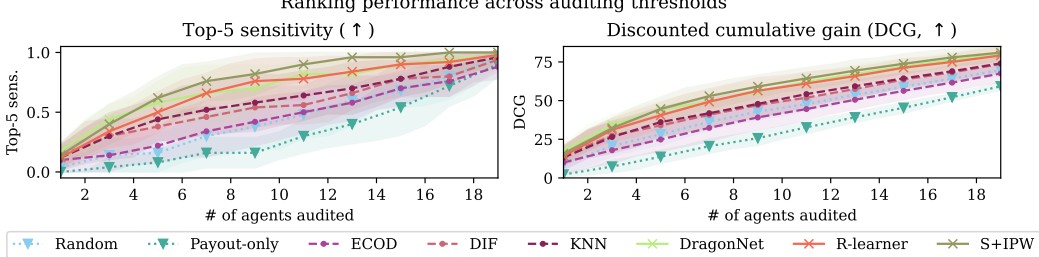

Figure 15: Mean top-5 sensitivity (left) and DCG (right) across # of agents audited at mean range 0.7, with $\pm\sigma$ error. $\nabla$: naïve baseline. $\circ$: anomaly detection method. $\times$: causal effect estimator.

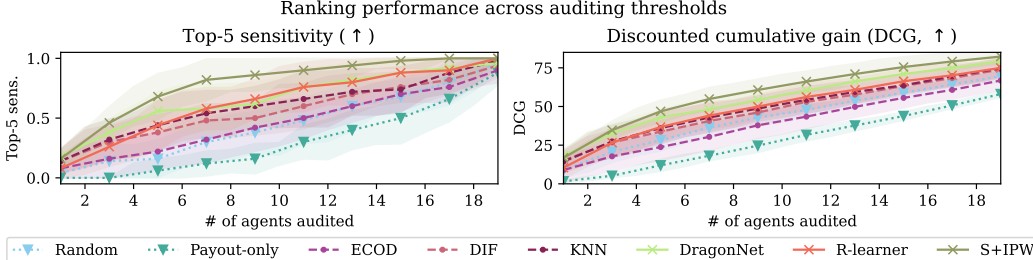

Figure 16: Mean top-5 sensitivity (left) and DCG (right) across # of agents audited at mean range 0.8, with $\pm\sigma$ error. $\nabla$: naïve baseline. $\circ$: anomaly detection method. $\times$: causal effect estimator.

- $R_\mu = 0.6$: Figure 14
- $R_\mu = 0.7$: Figure 15
- $R_\mu = 0.8$: Figure 16
- $R_\mu = 0.9$: Figure 17
- $R_\mu = 1.0$: Figure 18

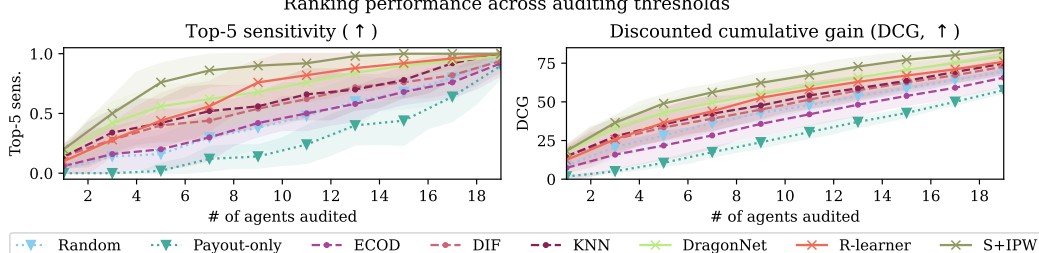

Figure 17: Mean top-5 sensitivity (left) and DCG (right) across # of agents audited at mean range 0.9, with $\pm\sigma$ error. $\nabla$: naïve baseline. $\circ$: anomaly detection method. $\times$: causal effect estimator.

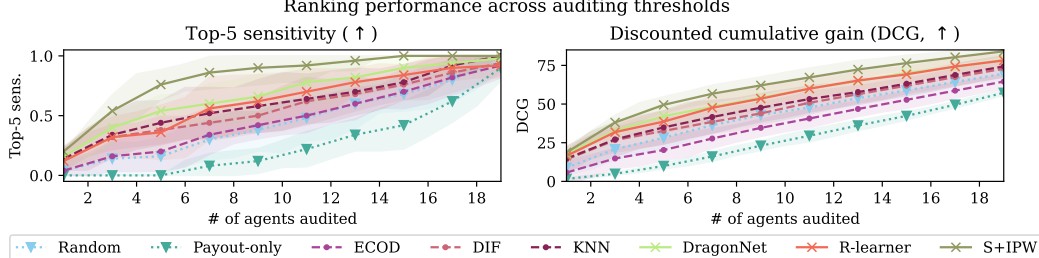

Figure 18: Mean top-5 sensitivity (left) and DCG (right) across # of agents audited at mean range 1.0, with $\pm\sigma$ error. $\nabla$: naïve baseline. $\circ$: anomaly detection method. $\times$: causal effect estimator.

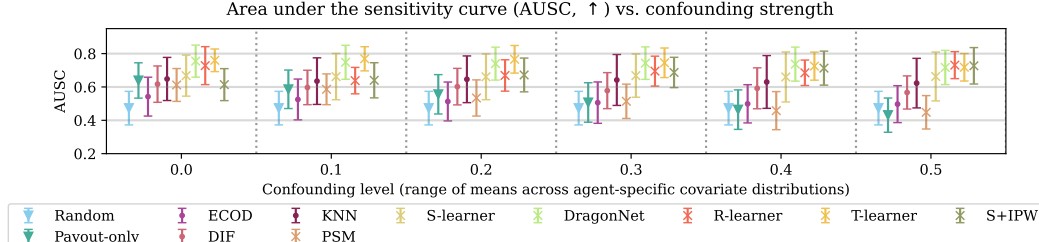

Figure 19: Area under the sensitivity curve (AUSC) for all methods tested across levels of confounding (mean range $R_\mu \leq 0.5$). $\nabla$: naïve baseline. $\circ$: anomaly detection method. $\times$: causal effect estimator.

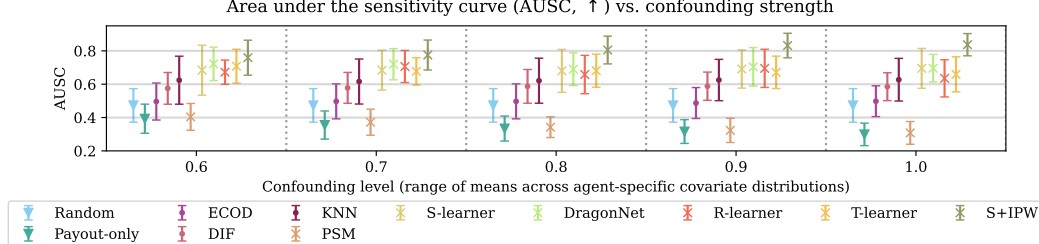

Figure 20: Area under the sensitivity curve (AUSC) for all methods tested across levels of confounding (mean range $R_\mu > 0.5$). $\nabla$: naïve baseline. $\circ$: anomaly detection method. $\times$: causal effect estimator.

We summarize the main trends for non-causal approaches and defer discussion of causal methods to the sensitivity analysis of all causal effect estimators. For convenience, we also plot the AUSC for *all* approaches across levels of confounding in Figures 19 ($R_\mu \leq 0.5$) and 20 ($R_\mu > 0.5$). **For readability, in contrast to our other figures, causal effect estimators are marked with "$\circ$" instead of "$\times$".**

**The payout-only approach can degrade to worse-than-random ranking due to confounding.** The random auditing method performs similarly across all levels of confounding, as expected.

However, as confounding increases, the payout-only ranking degrades toward random, then worse than random. The latter can occur if confounding is so strong that the relationship between gaming and observed diagnosis rates flips; *e.g.*, if (in the health insurance setting) very dishonest plans tend to serve relatively healthy populations compared to more gaming-averse plans.

**In the synthetic dataset, anomaly detection methods use the variance of the observed decision ($\mathbb{V}[d_i]$) as a gaming signature.** Most anomaly detection methods perform near-random, or slightly better than random. While this is due to the properties of the synthetic dataset, the results highlight potentially interesting characteristics of anomaly detection methods for gaming detection. Recall that the anomaly detection methods take covariates and agent decisions $(x_i, d_i)$ as input. $\mathbb{V}[x_i]$ is identical across agents by design, and all agents see the same number of observations. However, $\mathbb{V}[d_i]$ may vary across agents. Thus, KNN uses $\mathbb{V}[d_i]$ as a signal of gaming, which has utility under low confounding, but is less useful as confounding increases.

To see this, recall that $d_i$ is a binary decision, and let $p_d = P(d_i = 1)$ for some agent. $\mathbb{V}[d_i]$ is proportional to $p_d \cdot (1 - p_d)$, and is concave in $p_d$ with maximizer $p_d = 0.5$. Since $P(d_i)$ is generally low in simulation (*i.e.*, $\ll 0.5$), agents with higher observed $P(d_i)$ rates will generally have higher $\mathbb{V}[d_i]$ as well, yielding a higher anomaly score. Under no confounding, these are precisely the agents that are gaming more. Indeed, KNN performs slightly better than random, but its advantage over random performance diminishes slightly with confounding as the utility of $\mathbb{V}[d_i]$ as a signature for gaming. However, even at low confounding, KNN and related anomaly detection methods are inherently unable to detect gaming in non-outlier points, which occur in denser regions of covariate space. In these regions, causal methods enjoy an advantage over anomaly detection approaches due to improved overlap (Assumption 8). Thus, KNN does not exceed the ranking performance of the best causal methods. Note that this argument assumes that $\mathbb{V}[x_i]$ is similar across agents, which holds in the synthetic dataset.

More advanced anomaly detection methods can also achieve slightly better than random performance (*e.g.*, DIF), but since these approaches transform the covariate space in highly non-linear ways (*e.g.*, via random projections as in DIF, or via feature-wise CDFs as in ECOD), they may destroy outlier information useful for gaming detection. Ultimately, anomaly detection methods have inherently limited utility for gaming detection, since some gamed decisions may appear distributionally close to other gamed decisions.

## D.2 Sensitivity analysis of causal effect estimators

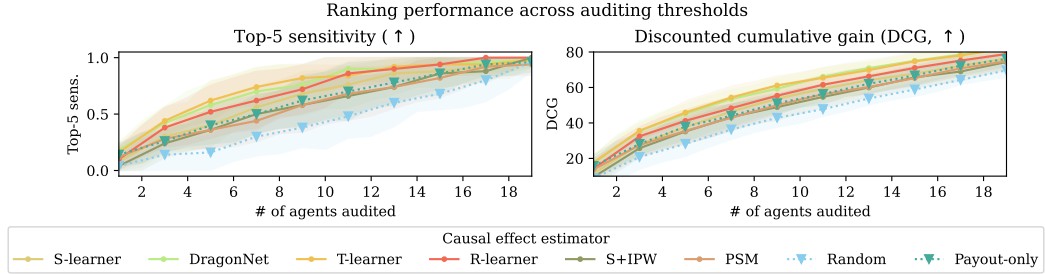

Figure 21: Sensitivity analysis of all causal methods tested, mean range 0.0. $\triangledown$: naïve baseline. $\circ$: causal effect estimator.

We show plots with the top-5 sensitivity and DCG at all levels of confounding evaluated (as measured by the mean range $R_\mu \in \{0, 0.1, \ldots, 1.0\}$) for causal effect estimators only, plus random and payout-only methods for comparison. An index of figures follows:

- Figure 21: $R_\mu = 0.0$
- Figure 22: $R_\mu = 0.1$
- Figure 23: $R_\mu = 0.2$
- Figure 24: $R_\mu = 0.3$
- Figure 25: $R_\mu = 0.4$

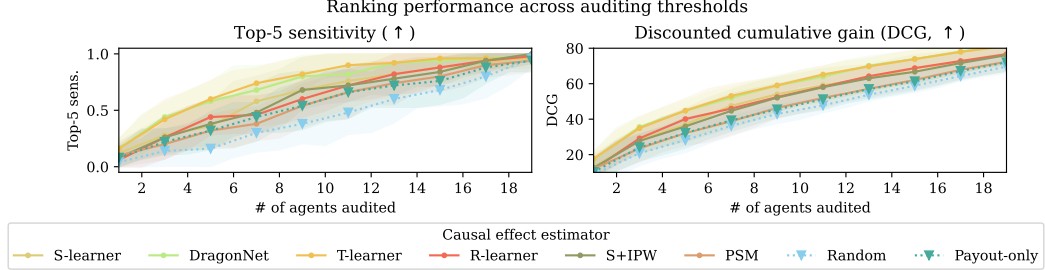

Figure 22: Sensitivity analysis of all causal methods tested, mean range 0.1. $\nabla$: naïve baseline. $\circ$: causal effect estimator.

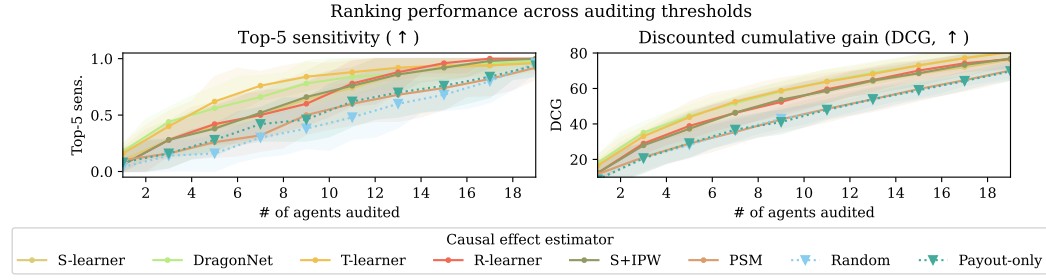

Figure 23: Sensitivity analysis of all causal methods tested, mean range 0.2. $\nabla$: naïve baseline. $\circ$: causal effect estimator.

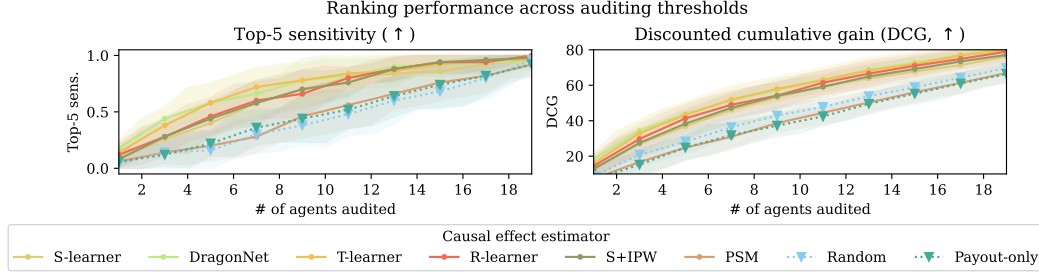

Figure 24: Sensitivity analysis of all causal methods tested, mean range 0.3. $\nabla$: naïve baseline. $\circ$: causal effect estimator.

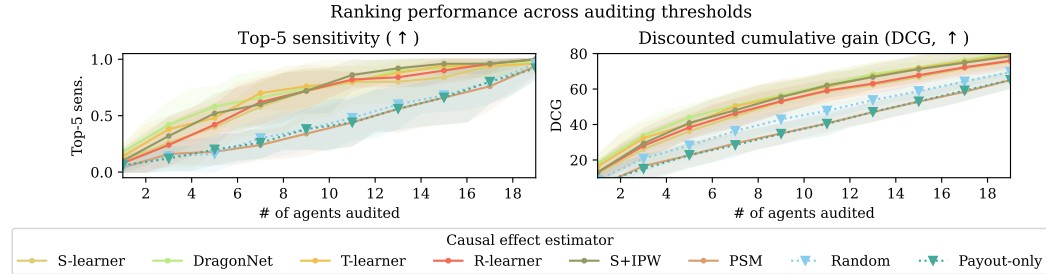

Figure 25: Sensitivity analysis of all causal methods tested, mean range 0.4. $\nabla$: naïve baseline. $\circ$: causal effect estimator.

- Figure 26: $R_\mu = 0.5$
- Figure 27: $R_\mu = 0.6$
- Figure 28: $R_\mu = 0.7$

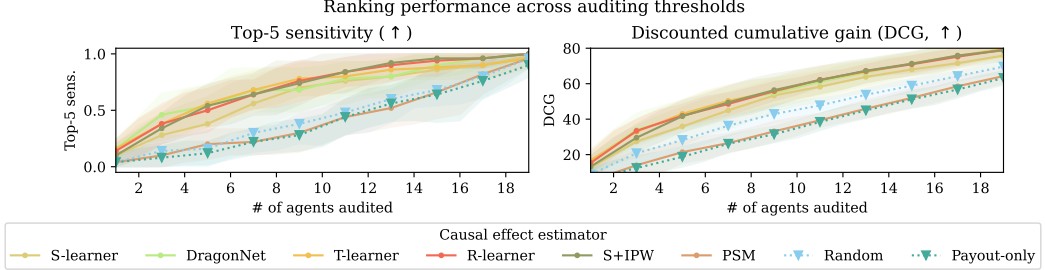

Figure 26: Sensitivity analysis of all causal methods tested, mean range 0.5. $\nabla$: naïve baseline. $\circ$: causal effect estimator.

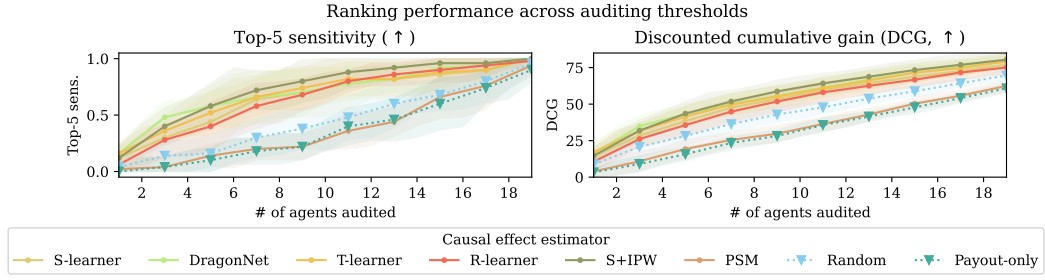

Figure 27: Sensitivity analysis of all causal methods tested, mean range 0.6. $\nabla$: naïve baseline. $\circ$: causal effect estimator.

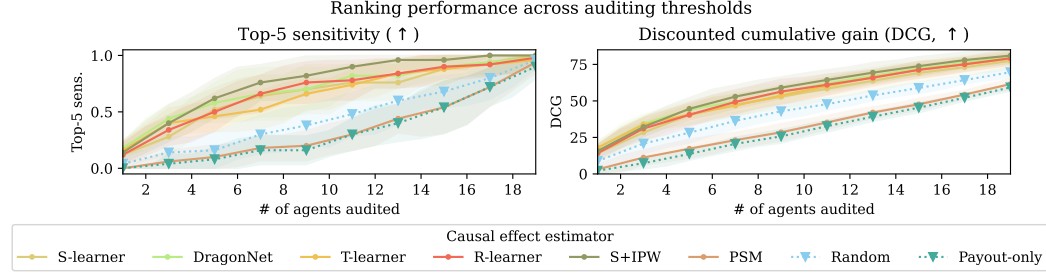

Figure 28: Sensitivity analysis of all causal methods tested, mean range 0.7. $\nabla$: naïve baseline. $\circ$: causal effect estimator.

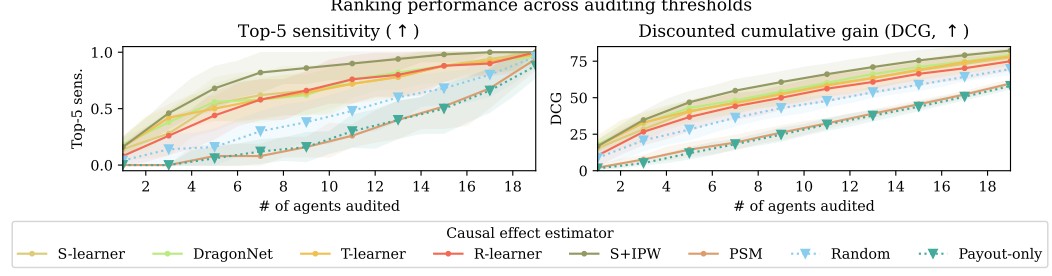

Figure 29: Sensitivity analysis of all causal methods tested, mean range 0.8. $\nabla$: naïve baseline. $\circ$: causal effect estimator.

- Figure 29: $R_\mu = 0.8$
- Figure 30: $R_\mu = 0.9$
- Figure 31: $R_\mu = 1.0$

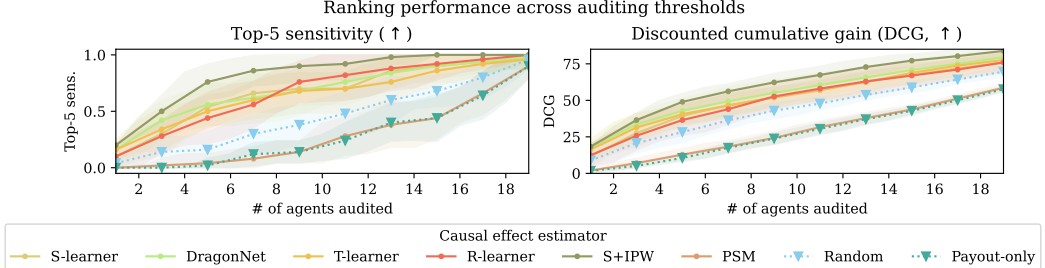

Figure 30: Sensitivity analysis of all causal methods tested, mean range 0.9. ▽: naïve baseline. ○: causal effect estimator.

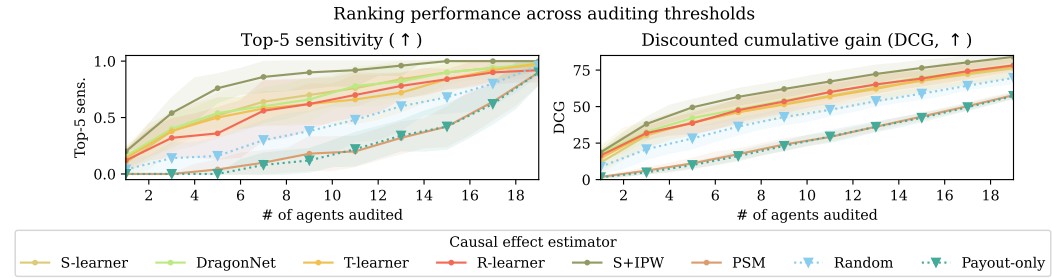

Figure 31: Sensitivity analysis of all causal methods tested, mean range 1.0. ▽: naïve baseline. ○: causal effect estimator.

Note that, even absent confounding, causal approaches outperform the payout-only model. This is because causal approaches explicitly incorporate the covariates into modeling, while the payout-only model directly uses the marginal outcome distribution. Incorporating covariates into regression models for treatment effect estimation can decrease estimator variance [37] (but is not guaranteed to do so), consistent with the empirical results.

Propensity score matching (PSM) also performs poorly across all levels of confounding. Since the matching approaches conduct matching pairwise across observations seen by agent pairs, the causal effect estimates are computed in a subset of similar observations across one pair of agents, but not the subset of similar observations across *all* distributions agent observations. This suggests that controlling for confounding simultaneously across all levels of treatment is potentially important for applying causal effect estimators to gaming detection. Ultimately, matching approaches may not scale to large numbers of treatments, such as those expected in multi-agent strategic adaptation. Non-optimal matching approaches (*e.g.*, greedy matching without replacement) are a potential workaround, but we leave the adaptation of matching methods to large numbers of treatments to future work.

We note that, at low levels of confounding, the difference between the S-learner and T-learner may be dataset-dependent: empirically, S-learners often regularize causal effect estimates towards zero, while T-learners thrive when causal effects are non-zero and heterogeneous [44, 55], as in our synthetic dataset. Thus, per-agent modeling, as done by the T-learner and DragonNet, can better capture the complex treatment effects in our dataset.

Furthermore, the R-learner and S+IPW both perform poorly at *low* levels of confounding, but improve at high levels of confounding. Since both the R-learner and S+IPW fit a nuisance propensity score estimator, this suggests that difficulties in propensity score estimation at low levels of confounding could potentially explain the observed trends.

The underperformance of the R-learner may be surprising given its doubly-robust properties, but the high-dimensional generalization [47] requires restrictive conditions for convergence. Formally, the R-learner fits four models $m, g, h, e$ of the form

$$d \sim m(x) + g(x)^\top (h(p) - e(x)), \tag{32}$$

where $m$ is fit independently, and $g, h, e$ are fitted using alternating optimization. The final treatment effect estimate of swapping from agent $p$ to $p'$ is given by $g(x)^\top h(p) - g(x)^\top h(p')$, but the oracle representation of $h(\cdot)$ is unknown and must be fitted. Convergence of the nuisance parameter estimate of $e(x)$ to the oracle value of $h(p)$ is necessary for convergence of the overall treatment effect estimate.

## E    Training

### E.1    Model architectures

All approaches that fit a model are built based on a fully-connected neural network with two hidden layers and 300 neurons per layer plus ReLU activations. The output of the neural network either has size two with a softmax non-linearity (for classification; *i.e.*, predicting agent decisions), or no activation and a pre-specified output size (*i.e.*, for generating feature maps in the high-dimensional R-learner).

We describe approach-specific modifications to the architectures as follows:

**S+IPW.** The estimated weights are used as a sample weight when training the S-learner. At inference time, weights are also computed for test examples based on the propensity model fitted on the training set to take an inverse propensity-weighted average of the S-learner estimates.

**DragonNet.** We changed the propensity prediction head from a binary classification (as in the original paper [45]) to a multi-class classification head, since there are multiple treatments in our setting. Furthermore, we introduce a new outcome modeling head per agent. Targeted regularization is omitted due to the multi-treatment setup.

**R-Learner.** Feature maps for all nuisance parameters have dimensionality 10. The generalized R-learner uses alternating optimization to fit some nuisance parameters, where two models representing a product decomposition of the response function are updated $K$ times for every update of a "propensity feature" model. We set $K = 10$; we defer to [47], page 6 for more details about the training procedure for the generalized R-learner, from which we designed our implementation.

### E.2    Anomaly detection hyperparameters

For KNN, we keep the neighborhood size at 5, the default value. DIF uses neural network-based random projections to compute an anomaly score. Thus, we use the same architecture for DIF as used for causal approaches, with an ensemble of 50 representations. Each representation is used as input into an isolation forest of size 6 [19].[8] ECOD does not take hyperparameters [22].

### E.3    Dataset splits

We use a seeded random development-test split (7:3) for all datasets. The development split is reserved for all model fitting, while all causal effect estimates are reported on the test split. The development split is further randomly split into a training and a validation set. All model selection techniques (*e.g.*, early stopping) are performed with respect to evaluation metrics on the validation set.

### E.4    Training hyperparameters

**Fully synthetic data**    We use the following hyperparameters for training all models:

- Optimizer: SGD with learning rate $10^{-2}$ and weight decay $10^{-3}$.

---

[8]An isolation forest is a separate anomaly detection method, in which the anomaly score is related to the number of random "splits" with respect to a single randomly-selected covariate (*i.e.*, a threshold of the form $x_i \geq r$) needed to isolate a point. This method assumes that outliers are "easier" to isolate, and require fewer splits. Multiple random-splitting routines (isolation trees) are ensembled to form an isolation forest.

- Learning rate schedule: We reduce the learning rate by a factor of 0.1 after 5 epochs of non-improvement with respect to the validation loss.
- Training length: A *maximum* of 1000 epochs, with early stopping (patience: 10 epochs) based on validation loss.

**Medicare FFS**

- Optimizer: SGD with learning rate $10^{-2}$ and weight decay $10^{-3}$.
- Learning rate schedule: We reduce the learning rate by a factor of 0.1 after 5 epochs of non-improvement with respect to the validation loss.
- Training length: A *maximum* of 1000 epochs, with early stopping (patience: 10 epochs) based on validation loss.

# F    Software and Hardware

## F.1    Software

All code was written in Python 3.10.4 (license: PSF). All non-causal anomaly detection approaches were implemented using PyOD (license: BSD 2-clause) [56]. All neural networks were implemented in PyTorch 2.2.0 (license: Custom "BSD-style"[9]) [57], using Skorch 0.15.0 (license: BSD 3-clause) [58] as a wrapper. Metrics were computed using both Scikit-Learn 1.3.2 (license: BSD 3-clause) [59] and Scipy 1.11.4 (license: BSD 3-clause) [60]. For the fully synthetic data generation process, CVXPY 1.4.2 (license: Apache 2.0) [61] was used to solve each agent's utility maximization problem, and used in tandem with SCIP 9.0 (`pyscipopt` 5.0.0; license: Apache 2.0) for the matching approaches (formulated as mixed-integer programs) [62]. Numpy 1.22.3 (license: BSD-style) [63][10] and Pandas 2.0.3 (license: BSD 3-clause) [64] were used for data manipulation. Matplotlib 3.8.2 (empirical results; license: PSF-style)[11] and Adobe Illustrator 2023 (overview figures; license: commercial, "Named User Licensing"[12]) were used for figure generation. For the Medicare cohorts, we generated HCC (Hierarchical Condition Categories; used by the Center for Medicare Services) codes from raw diagnosis codes reported in claims data via HCCPy 0.1.9 (license: Apache 2.0)[13].

Other dependencies include `tqdm` 4.66.2 for rendering progress bars (license: MPL 2.0 and MIT), `gitpython` 3.1.43 for bookkeeping (license: BSD 3-clause), `pandarallel` 1.6.5 for parallel data processing (license: BSD 3-clause), and `ruamel` 0.18.6 (license: MIT) for configuration file management. All software excepting Adobe Illustrator is open-source and free for use.

## F.2    Hardware

All experiments were run on either one Titan V or V100 GPU using 12.9GB of RAM as managed via a Slurm job submission system. Computing nodes had two 2.10GHz Intel Broadwell (Xeon E5-2620V4) processors each (16 cores total). Execution time was limited to six hours per run, but all training runs (one model type on 10 datasets) lasted under one hour due to the relatively small size of the architectures and datasets under consideration.

# G    Code

The fully-synthetic datasets, experimental code, and implementations of all approaches under evaluation will be made publicly available at https://github.com/MLD3/gaming_detection. The authors do not have permission to release any of the Medicare data, but will release the relevant data processing code.

---

[9]See https://github.com/pytorch/pytorch/blob/main/LICENSE.
[10]See https://github.com/numpy/numpy/blob/main/LICENSE.txt.
[11]See https://matplotlib.org/stable/project/license.html.
[12]See https://helpx.adobe.com/enterprise/using/licensing.html.
[13]https://github.com/yubin-park/hccpy

