# OpenReview forum: "Who’s Gaming the System? A Causally-Motivated Approach for Detecting Strategic Adaptation"
_NeurIPS.cc/2024/Conference — NeurIPS 2024 poster_

### Official Review · Reviewer_Drhf · 2024-07-11

**Soundness:** 3
**Presentation:** 3
**Contribution:** 3
**Rating:** 6
**Confidence:** 2

**Summary:**

The paper studied how to identify and rank agents who strategically manipulate their inputs to game machine learning models in multi-agent settings. A causally-motivated approach was proposed to address this challenge.

**Strengths:**

The paper is an interesting follow-up to [1]. Provided that the proposed mythology is solid, I can foresee many scenarios where it can be applied. Especially, I feel that it may be applied to analyze college admission mechanisms.

[1] Hardt, Moritz, et al. "Strategic classification." Proceedings of the 2016 ACM conference on innovations in theoretical computer science. 2016.

**Weaknesses:**

The paper assumes players to be almost perfectly rational, i.e., they have a strong motivation to maximize utility. In reality, however, people are often only bounded rational. If so, that is, if some players do not particularly care about their utility, how would the accuracy of your approach be affected?

**Questions:**

Please find my first question in the above part. Here's another question.

After the proposed approach is adopted and the player gaming the most is identified, how can we find our if the identification is correct? That is to say, how to validate your approach.

**Limitations:**

The authors have appropriately discussed the limitations.

---

> ### Author Rebuttal · Authors · 2024-08-06
>
> We thank Reviewer Drhf for their comments. We appreciate that the reviewer found our approach to be practical, and applicable to other real-world problems such as college admissions mechanisms.
>
> **Real-world validation.** Great point — real-world validation is an inherent limitation of the synthetic data setting indeed. To mitigate this issue, we did aim to verify that our rankings line up with known drivers of upcoding, such as the prevalence of private healthcare providers in a state as studied in [C1, C2] (Section 5.2; also see Table 1, pg. 9). In practice, real-world validation of causally-motivated approaches, including ours, is difficult, but this limitation is shared by many works in causal inference [C3, C4] that also opt for synthetic/semi-synthetic evaluations [C5] to verify the theoretical claims.
>
> Furthermore, we envision our ranking-oriented approach as an initial flag for auditors to subsequently investigate and confirm/rule out gaming.  A purely algorithmic validation grounded in the observed data is likely infeasible under our problem assumptions as per our Proposition 1, since any uncertainty in the correct ground truth decisions results in uncertainty in the gaming parameter.
>
> **Robustness to the rational actor assumption.** This is an insightful point. Indeed, while rationality is a standard assumption, and much game-theoretic literature shares this limitation, we will more explicitly highlight the assumption that agents are rational as a limitation. However, we wanted to highlight that our focus on rankings of gaming propensity rather than point estimates of the gaming parameter could afford some robustness via Proposition 2 (Section 3.2, pg. 5).
>
> Intuitively, violations of the rational actor assumption could simply be another source of noise; then, if such violations plus estimation error are too small to flip any pair of rankings with respect to the ground truth, then our estimated gaming rankings will still match the ground-truth rankings. Informally stated:
>
> Let $\Delta_p(d^*_p)$ be the optimal utility-maximizing response, and let $\tilde{\Delta}_p(d^*_p)$ be the observed response. If $|\Delta_p(d^*_p) - \tilde{\Delta}_p(d^*_p)| \leq \varepsilon_1$ and $\sup |\tau - \hat{\tau}| \leq \varepsilon_2$ for some $\varepsilon_1, \varepsilon_2 > 0$, then no rankings where the ground-truth $|\hat{\tau}(p, p’)| > \varepsilon_1 + \varepsilon_2$ will be flipped.
>
> Note that this corollary requires stronger conditions than Proposition 2, which only accounted for estimation error in the causal effect estimator $\hat{\tau}$.
>
> We conclude by grounding our discussion in the motivating example of gaming the CMS-HCC model. Here, the notion of rationality is well-defined: rational agents game (i.e., perturb their reported diagnoses) such that profit is maximized; our utility function directly corresponds to a dollar amount. We speculate that deviations from the rationality assumption could occur if such agents are unable to carry out the utility-maximizing action (i.e., insufficient resources to, for example, schedule a home visit with a healthcare professional to generate diagnosis codes [C6]), or incorrectly estimate the utility-maximizing action (e.g., agents may incorrectly estimate costs). We leave the formalization of these rationality violations to future work. We will add this discussion of potential violations of the rational actor assumption and its implications on our ranking-based framework to Appendix B.4.
>
> **References**\
> [C1] Silverman, Elaine, and Jonathan Skinner. "Medicare upcoding and hospital ownership." Journal of health economics 23.2 (2004): 369-389.\
> [C2] Silverman, Elaine, and Jonathan S. Skinner. "Are for-profit hospitals really different? Medicare upcoding and market structure." (2001).\
> [C3] Shi, Claudia, David Blei, and Victor Veitch. "Adapting neural networks for the estimation of treatment effects." Advances in neural information processing systems 32 (2019).\
> [C4] Louizos, Christos, et al. "Causal effect inference with deep latent-variable models." Advances in neural information processing systems 30 (2017).\
> [C5] Hill, Jennifer L. "Bayesian nonparametric modeling for causal inference." Journal of Computational and Graphical Statistics 20.1 (2011): 217-240.\
> [C6] Weaver, C., et al. “Insurers Pocketed $50 Billion From Medicare for Diseases No Doctor Treated.” The Wall Street Journal, 8 July 2024. https://www.wsj.com/health/healthcare/medicare-health-insurance-diagnosis-payments-b4d99a5d

---

### Official Review · Reviewer_P2tW · 2024-07-12

**Soundness:** 4
**Presentation:** 2
**Contribution:** 3
**Rating:** 6
**Confidence:** 2

**Summary:**

This work studies the problem of identifying agents who would likely game a given system. When the gaming parameters are unknown, the authors show that identifying these parameters requires strong assumptions. In contrast, they show an ordering of agents based on their ranking order is learnable from a dataset. They use this ranking to detect gaming in a Medicare application.

**Strengths:**

The authors generalize the study of strategic adaptation to a much more realistic setting and provide provable results for computing rankings, which can then be used to detect gaming. The application of Medicare is also interesting.

**Weaknesses:**

Although the framework is interesting, it is unclear in general how to use rankings to detect gaming, which is the motivator of the study. A general provable approach to using rankings for gaming, even under some conditions, would improve the applicability of this work.

**Questions:**

Is there any general approach to using rankings as a subroutine to detect gaming? If so, under what conditions would this approach provably work?

**Limitations:**

Yes.

---

> ### Author Rebuttal · Authors · 2024-08-06
>
> We thank Reviewer P2tW for their comments. We appreciate that the reviewer found the application area of U.S. Medicare to be interesting.
>
> **Can we actually detect gaming/use it as a subroutine?** Good question — to that end, our Proposition 1 demonstrates that definitive gaming detection is not possible without knowing ground-truth decisions. Intuitively, if the ground truth “correct” decisions were known/easy to collect, gaming detection would be trivial: we could simply check all observed decisions against ground truth in such cases. In summary:
> * **No framework under our assumptions can answer (via Proposition 1):** “Is Agent 1 gaming their decisions?”
> * **Our framework can answer (via Theorem 1 & Corollary 1):** “Is Agent 1 gaming their decisions more/less than Agent 2?”
> Under our assumptions, to enable definitive gaming detection, one would need to assume that ground truth is fully observed, which is likely too strong of an assumption (i.e., everyone is subjected to an accurate audit, or is honest about whether they gamed). As-is, we envision that our ranking-oriented approach would serve as an initial flag for auditors to prioritize which agents to investigate. So, one could think of our ranking-based gaming detection approach could be a “subroutine” in a policy/procedural sense (albeit not in a purely algorithmic sense).
>
> However, the reviewer’s comment motivated us to revisit Proposition 1 and explore whether tighter versions of this bound could assist in gaming detection. We found a simple extension of our existing results that could allow for partial gaming detection. Let’s assume, for sake of exposition, that we can evaluate $R’$ and $c’$, similarly to the two-agent example of Section 3.1. This means we can evaluate the lower-bound of Proposition 1. Then, a partial gaming detection approach might take on two steps:
> 1. Define a threshold $\lambda^*$ such that plans with $\lambda_p$ lower-bound greater than $\lambda^*$ are definitively not gaming
> 2. Estimate upper and lower bounds on $d^*_p$ to yield tighter lower/upper bounds on $\lambda_p$; plans with $\lambda_p$ upper-bound less than $\lambda^*$ are definitively gaming.
>
> Such a threshold might be defined via some tolerance for deviation from ground truth; *i.e.*, we can define gaming as $\Delta_p(d^*_p) - d^*_p \geq \varepsilon$ for some $\varepsilon > 0$, which allows agents to have an error rate of $\varepsilon$ without being considered to be gaming. Based on $\varepsilon$, we can then compute a “threshold” value of $\lambda^*$ that demarcates gaming from non-gaming agents. For the two-agent example in Section 3.1 where $R(x) = x$ and $c(x) = x^2$, this would yield $\lambda^* = \frac{1}{2\varepsilon}$.
>
> Then, we can combine the $\lambda^*$ threshold with a tighter version of the bounds of Proposition 1. Suppose that we can claim $d^p \in [\underline{d}^*_p, \bar{d}^*_p]$. Then, revisiting Eq. 8 (Appendix B.1, pg. 14) of our proof for Proposition 1, we’d reach the following:
>
> $$\lambda_p \in \left( \frac{R’(\Delta(d^*_p))}{c’(\Delta(d^*_p), \bar{d}^*_p)}, \frac{R’(\Delta(d^*_p))}{c’(\Delta(d^*_p) - \underline{d}^*_p)} \right).$$
>
> For clarity, let’s abbreviate the terms in these bounds as $\underline{b}$ and $\bar{b}$ for the lower and upper bounds, respectively, such that $\lambda_p \in [\underline{b}, \bar{b}]$. Since we assumed $R’$ and $c’$ can be computed, and we assume estimates of $\bar{d}^*_p$ and $\underline{d}^*_p$ are available, we can calculate $\underline{b}$ and $\bar{b}$ directly. Thus, we would conclude:
> * If $\bar{b} \leq \lambda^*$, then the agent is gaming (where $\lambda^*$ depends on the error tolerance $\varepsilon$), and
> * If $\underline{b} \geq \lambda^*$, the agent cannot possibly be gaming.
>
> Although such a method would be unable to detect or rule out gaming in agents for which $\lambda^* \in [\underline{b}, \bar{b}]$, it could provide more definitive gaming detection results for a subset of agents.
>
> Note that this idea is preliminary — we have not had the opportunity to empirically validate this or identify similar approaches in the literature. This method also introduces two previously unused assumptions:
> 1. One can compute $R’$ and $c’$
> 2. One can estimate lower and upper bounds on $d^*_p$.
>
> While prevalence estimation via a random sample could yield confidence interval-based bounds on $d^*_p$, and $R’$ can be calculated if we know the model to be manipulated (e.g., the publicly-available CMS-HCC model used by U.S. Medicare), evaluating $c’$ may be unrealistic since it can be any strictly-convex function. Although testing and proposing such a new approach is likely out of scope for our revision, we will refine this argument and comment on this direction in Appendix B.1 as a potential avenue of future work.

---

> > ### Comment · Reviewer_P2tW · 2024-08-12
> > **Rebuttal Response**
> >
> > Thank you for your response! Although the ranking approach is interesting, I will stick with my given scores since the main goal of the paper is to detect gaming. If your preliminary idea works out, it would greatly strengthen the paper.

---

### Official Review · Reviewer_dPj8 · 2024-07-13

**Soundness:** 2
**Presentation:** 2
**Contribution:** 2
**Rating:** 3
**Confidence:** 3

**Summary:**

The paper considers the problem of identifying agents with the highest values of scaling parameters in a stylized strategic adaptation optimization model under a wide range of assumptions. The paper casts this problem as a ranking problem via causal effect estimation and provides an algorithm to rank the parameters of agents.

The paper then provides an experimental evaluation using synthetic and real-world medicare data to show the observations from the proposed models and algorithm.

**Strengths:**

+ The paper considers an interesting manipulation problem

**Weaknesses:**

- The motivation and justification for the considered problems seem to be lacking at the beginning (mostly presented with very little context and connections)

- The model is very marginal compared to existing studies (i.e., the new component seems to assume non-similarities in gaming via a cost-scaling factor); the notations/paper contents are poorly written, the results have few explanations in terms of implications, and there are assumptions in the models that are hard to justify

- The proposed approaches (i.e., casts as causal effect estimation) have little to no justifications

- The synthetic experimental evaluation is based on a single dataset with manual tuning that has little to no explanations, which limits the generalizability of the observations

**Questions:**

N/A

**Limitations:**

See the above and below comments.


Abstract

"machine learning models" -> provide examples of how the interaction would look like

"their inputs to the model" -> such as? what are the outcomes here? from the ML models? what utility? the agents using the models for something? it is very unclear

"We consider a multi-agent setting" -> what is the motivation for this goal? what is the implication if it is achieved? why do we care if it games? What do you mean by aggressively? in fact, how do you even define aggressive?

"identifying such agents is difficult" -> why? it is not clear to me how would you do this even if you know the utility; why make it more difficult now?

" is parameterized via a scalar" -> the meaning is not clear; what is parametrized? agent utility function? what is a scalar here? why is this realistic at all?

"is only partially identifiable" -> what does it mean by this?

"By recasting the problem" -> why recasting the problem? why this approach is justifiable?
"causal effect estimation problem" -> what is this problem? what is the connection to the worst offenders? very confusing here; the next phrase makes little to no sense at all; why/what is the identifiable property?

"in a synthetic data study" -> only one data set?
what is this coding behavior? what does it have to do with gaming?
what are the features? what do you mean associated with gaming?

1 Introduction

"guide decisions that impact individuals or entities" -> provide examples

please review the literature on mechanism design and game theory as well as their ML related applications

"obtain a more desirable outcome" -> outcome is what here? what is the connection between ML models and outcomes?

"to the difficulty of generating supporting evidence" -> provide examples; explain more

""utility maximization:" -> fix

"to maximize a payout" -> what is the payout? utility function?

"which calculates a" -> the government or the companies?
" via a publicly available model" -> how do you know they use this model?

"Companies may attempt to" -> what happens to the companies that do this?

"Beyond health insurance, gaming emerges" -> how does health insurance impact individuals? who is doing the gaming in the said applications? what are you trying to change in the features?

Also, are the models already trained? or are you talking during learning?

"agents with the highest propensity " -> how do you measure this?

"given a dataset of agents" -> dataset of agents doesn't sound right? what does it mean here? do you mean a set of agents?
"their observed model inputs" -> is the model fixed here? how do they interact with the models? why can't do this together?

"fraud/gaming labels" -> what does this mean?

what is the utility here? what are they affecting? what is the connection to distributional assumptions? these are used with little to no context

"But past works in strategic adaptation" -> what does it mean by costs in this context? is that the only difference from this vs the other models?

"scalar gaming parameter that scales costs " -> the meaning of this is not clear; what is this parameter? it is not explained in texts; the figure is not meaningful without a clear description

"partially identifiable" -> what is the meaning of this?

"However, by recasting gaming detection as causal" -> again, why is this justifiable? what is this causal effect estimation going to do?

"that ranking agents" -> what ranking is solving? I am very confused

"a cost-scaling factor" -> why is this more realistic?

"Furthermore, much work in strategic a" -> this paragraph seems to be more like related work section; it is out of place and disrupts the flow of reading; you should probably consider an independent related work section; they are also hard to understand with little to no background context presented

What is the contribution here? there should be a contribution subheadings and/or related work also

"a synthetic dataset" -> how do you have a ground truth on this? what about other datasets?

"causal approaches rank the " -> hard to understand this sentence

"healthcare providers, a suspected driver of gaming" -> you can say that unless you have concrete evidence; otherwise, you will sued with claims like this

"In summary: we " -> this paragraph provides new information that is not discussed or connected to earlier messages


2 Background & Problem Setup

"to a payout" -> what is a payout here? why f is mapping only a single agent attribute? should it maps from more agents? or even datasets?

"according to some function" -> what is this function? How is R dependent on d and f? why d' has to be in D?

"For simplicity, we assume R = f ; " -> if R = f, the function in (1) makes no sense what is the meaning of f has itself as input? the math is not correct

How is strategic classification used in strategic adaptation?


"To extend strategic adaptation to multiple agents" -> why is this justifiable at all?

What is M_p? What is d_i modeling? what decisions are they making?

"Agent assignment is" -> how is this model? how do you indicate assumption; i am still not clear about D_p? is that for an agency? within the agnecy they have M_p agents who can manipulate?

" to obtain a higher payout." -> why do they want to increase d_i? why can't they decrease?

What do they perturb the average instead of individual d_i? how does this connect to f defined earlier?

"is the ground truth value" -> how do you actually get the ground truth? before you define c(,) as two parameters and now you have a single parameter; it doesn't look consistent

"we introduce assumptions on the" -> you need to justify these assumptions; are they common in the literature? how do you model multiple agents here? it seems (2) only for one agent

3 Theoretical analysis: finding agents most likely to game

"We aim to identify agents most likely" -> wait; is lambda_p unknown? why is this the right way to identify agents?

"be point-identified" -> unclear meaning
what is the meaning of partially identifiable? You need to provide implications of Prop 1

I still don't get why  estimating counterfactuals is used in this context; how come figure 2 has no in-text explanations?

There are algorithms and figure on page 5 without explanations and connections to the paper


4 Empirical results & discussion


"We hand-select " -> what about other connection

---

> ### Author Rebuttal · Authors · 2024-08-06
>
> We thank Reviewer dPj8 for their detailed comments, which improved our work. First, we address the motivation for our approach and clarify conceptual questions. We then briefly discuss connections to past work. Then, we address concerns about generalizability and our assumptions. We will add these clarifications to our revision.
>
> **Motivation for causally-motivated/counterfactual-based approach.** Our main contribution is providing theoretical and empirical evidence that causal effect estimation is a well-motivated, practical avenue for ranking agents by their gaming parameter (i.e., propensity to game). To see why causal inference is needed, consider the following: we observe two insurance companies/healthcare providers, one of whom receives a much higher payout due to higher reported rates of a certain condition. That provider could be seeing sicker patients or they could be gaming and inflating reported diagnosis rates to secure the higher payout. Causal inference methods help us adjust for underlying patient health and compare these reported rates. If the two patient populations are similarly healthy, yet one provider receives a higher payout, it indicates potential gaming/fraud.
>
> Concretely, we use pairwise comparisons between agents to construct a ranking of agents by their gaming parameter. Theorem 1 shows that this pairwise comparison problem is equivalent to a causal effect estimation problem (i.e., estimating counterfactual decisions under counterfactual agents) under our assumptions. This yields the algorithm in Figure 4.
>
> **Why ranking?** Given a ranking, one could prioritize agents to screen/audit for gaming. One of our main results is demonstrating that directly predicting gaming is impossible without access to ground truth labels stating which agents are gaming. Proposition 1 shows that, absent stronger assumptions, we can only estimate a loose lower bound on the gaming parameter. To reinforce this result, we provide a two-agent example in Section 3.1 (L126-L132) showing how this lower bound may flip rankings with respect to the ground truth gaming parameter. While directly estimating tendency to game is impossible, we show that recovering a ranking is possible.
>
> **Other conceptual questions.** We provide clarifications about our approach not covered above.
> * **Q:** We group these questions about the gaming parameter $\lambda_p$:
>     - "We aim to identify agents most likely" -> wait; is lambda_p unknown? why is this the right way to identify agents?
>     - "agents with the highest propensity " -> how do you measure this?
>     - "scalar gaming parameter that scales costs " -> the meaning of this is not clear; what is this parameter? it is not explained in texts
> * **A:** Yes, $\lambda_p$ is an unknown scalar, and is how we define the propensity to game. One can think of it as a “penalty factor” for gaming. Lower values of $\lambda_p$ indicate higher gaming propensity, since such $\lambda_p$ down-weights the cost of gaming. We aim to identify agents who are most likely to game (lowest $\lambda_p$). This is of interest in settings where one has limited resources to audit gaming. A ranking approach allows auditors to prioritize investigating a subset of agents in which gaming is more likely (e.g., the top-$K$ agents with the lowest gaming parameters).
> * **Q:** "fraud/gaming labels" -> what does this mean?
> * **A:** Fraud/gaming labels refer to ground-truth indicators that denote whether an agent is definitively gaming. With such labels, one could simply use supervised learning to detect gaming, but we assume no such labels in our setting.
> * **Q:** "identifying such agents is difficult" -> why? it is not clear to me how would you do this even if you know the utility
> * **A:** If we know the *value* of the utility function and the reward function (i.e., in cases when the reward model $R$ is exactly the model to be manipulated $f$, as in Section 2), then we can infer that cost = utility - reward. Then, all agents that incurred non-zero cost (due to manipulating their own features) gamed by definition. We will more clearly delineate this in the revision.
> * **Q:** "obtain a more desirable outcome" -> outcome is what here? what is the connection between ML models and outcomes?
> * **A:** The “desirable outcome” is obtaining higher payout (e.g., higher compensation for treating patients with more complicated conditions). Such an ML model ($f$ in Section 2.1) might take as input the conditions that a provider claimed to treat, and output a dollar payout amount (e.g., via the U.S. government’s CMS-HCC model [A1]).
> * **Q:** “Also, are the models already trained? or are you talking [about] during learning?”
> * **A:** We assume a fixed model. Agents are responding to that model by perturbing their own features according to a utility-maximization problem. We make no further assumptions on the model training/fitting procedure.
> * **Q:** "to obtain a higher payout." -> why do they want to increase d_i? why can't they decrease?
> * **A:** Good question: since agents are utility-maximizing by assumption, decreasing $d_i$ can provably never maximize utility. We comment on this in Remark 1 (L106-107): note that costs are zero for $\Delta_p(d^*_p) < d^*_p$ (Assumption 3), but the reward function is increasing; hence, decreasing $\Delta_p(d^*_p)$ below ground truth reduces utility.
> * **Q:** "partially identifiable" -> what is the meaning of this?
> * **A:** Partial identifiability (e.g., Definition 18.1, [A2]) denotes that multiple parameter estimates are compatible with the data; e.g., a range of values of $\lambda_p$ as in our Proposition 1.

---

> ### Author Response · Authors · 2024-08-06
> **Rebuttal by Authors (part 2/2)**
>
> **Connections to past work.** The reviewer suggests connections to the mechanism design literature. We disagree: our proposed framework differs from mechanism design. Algorithmic mechanism design aims to maximize some social surplus (e.g., sum of utilities for all agents, in the language of our framework) by learning an optimal allocation rule (e.g., what we call a “model”) of some “resource” to agents [A3]. In contrast, our model/allocation rule is fixed, and we study how multiple individual agents respond. This aligns with our motivating problem setting in U.S. Medicare: given a fixed model (CMS-HCC) for calculating payouts to healthcare providers/insurance companies, providers/companies adjust their behavior accordingly. A mechanism design approach could inform upgrades to the CMS-HCC model, but is not our focus.
>
> We believe that our review of game-theoretic work in machine learning is extensive, though we are happy to consider suggestions to discuss specific papers. To that end, the closest related area is strategic classification, in which agents maximize their utility function in response to an ML model’s decisions. To that end, we have provided citations to related settings in L34-L38. We pay special attention to works with similar assumptions to ours, *e.g.*:
> * Utility functions are partially known (their setting) [A4] vs. utility functions are unknown beyond the general form (our setting)
> * Multiple agents with different capacities to game, modeled by norm constraints on manipulation (their setting) [A5] vs. different cost-scaling factors (our setting)
>
> We will consider a separate section/heading for the related works.
>
> **Model is “very marginal.”** We disagree that this is a weakness: small changes to the problem formulation may significantly affect the solution space. However, in addition to a novel utility-maximization formulation for gaming in multiple agents, we contribute theoretical analyses of the resultant utility-maximization problem, which motivates a causal effect estimation approach to producing a ranking of agents by gaming propensity. To our knowledge, our approach is the first to adopt a causal effect estimation approach to gaming ranking (though previous works have explored causal mechanism design approaches to counter gaming [A6, A7]).
>
> **Evaluation on one synthetic dataset => limited generalizability?** We partially disagree: a synthetic dataset for evaluation is necessary to establish proof-of-concept of the proposed approach. This is because ground truth gaming labels are inherently difficult/infeasible to obtain in practice (e.g., require accurate audits of all agents, or accurate self-reporting of fraud). Synthetic data validation is standard in causal inference (e.g., [A8, A9] are two well-established causal effect estimators that use IHDP [A10], a synthetic dataset, for validation).
>
> To further mitigate generalizability issues, we verify our rankings align with known drivers of upcoding (e.g., the state-level prevalence of private healthcare providers [A11, A12], Section 4.3, Table 1, pg. 9). We also align our design choices with our problem assumptions (Appendix C.1, pg. 16-17). We are happy to consider specific suggestions for other datasets or design choices that could improve the generalizability of our findings.
>
> **Are assumptions common in the literature? How are they justified?** We justify our assumptions after their introduction on L92-110, with examples where applicable. We like the suggestion to discuss how common our assumptions are in the literature; in particular, many strategic classification works implicitly use some form of Assumption 1 [A6, A13], and our Assumptions 2-4 are more general cases of assumptions found in [A4, A13, A14]. Assumption 5 ensures that the ground truth is predictable from the observed variables $x_i$, and is related to assumptions of no unmeasured confounding [A15]. Assumptions 6-8 are standard assumptions in the causal inference literature (e.g., Chapter 3 of [A15]). These hold independently for all agents. We will add these points to the revision.
>
> **Other changes.** Thank you for your diligence in checking our notation — we appreciate the careful notes! We will double-check that all variables are well-defined and identify places where we’ve used similar notation/overloaded notation.

---

> ### Author Response · Authors · 2024-08-06
> **References cited in rebuttal to Reviewer dPj8**
>
> **References**\
> [A1] Report to Congress: Risk Adjustment in Medicare Advantage, December 2021. https://www.cms.gov/files/document/report-congress-risk-adjustment-medicare-advantage-december-2021.pdf\
> [A2] Ding, Peng. A first course in causal inference. CRC Press, 2024.\
> [A3] Roughgarden, Tim. Twenty lectures on algorithmic game theory. Cambridge University Press, 2016.\
> [A4] Dong, Jinshuo, et al. "Strategic classification from revealed preferences." Proceedings of the 2018 ACM Conference on Economics and Computation. 2018.\
> [A5] Shao, Han, Avrim Blum, and Omar Montasser. "Strategic classification under unknown personalized manipulation." Advances in Neural Information Processing Systems 36 (2024).\
> [A6] Bechavod, Yahav, et al. "Gaming helps! learning from strategic interactions in natural dynamics." International Conference on Artificial Intelligence and Statistics. PMLR, 2021.\
> [A7] Horowitz, Guy, and Nir Rosenfeld. "Causal strategic classification: A tale of two shifts." International Conference on Machine Learning. PMLR, 2023.\
> [A8] Shi, Claudia, David Blei, and Victor Veitch. "Adapting neural networks for the estimation of treatment effects." Advances in neural information processing systems 32 (2019).\
> [A9] Louizos, Christos, et al. "Causal effect inference with deep latent-variable models." Advances in neural information processing systems 30 (2017).\
> [A10] Hill, Jennifer L. "Bayesian nonparametric modeling for causal inference." Journal of Computational and Graphical Statistics 20.1 (2011): 217-240.\
> [A11] Silverman, Elaine, and Jonathan Skinner. "Medicare upcoding and hospital ownership." Journal of health economics 23.2 (2004): 369-389.\
> [A12] Silverman, Elaine, and Jonathan S. Skinner. "Are for-profit hospitals really different? Medicare upcoding and market structure." (2001).\
> [A13] Hardt, Moritz, et al. "Strategic classification." Proceedings of the 2016 ACM conference on innovations in theoretical computer science. 2016.\
> [A14] Levanon, Sagi, and Nir Rosenfeld. "Strategic classification made practical." International Conference on Machine Learning. PMLR, 2021.\
> [A15] Robins, James, and Hernan, Miguel A. “Causal Inference: What If?” Boca Raton: Chapman & Hall/CRC. (2020).

---

### Author Rebuttal · Authors · 2024-08-06

We thank the reviewers for their insightful comments, which helped improve our work. Reviewers positively commented on the interestingness of our problem setting [R1, R3], as well as the practicality and realism of our proposed methodology [R2/R3]. We respond to requests for clarification from reviewers individually.

---

### Decision · Program_Chairs · 2024-09-25

**Decision:**

Accept (poster)

**Comment:**

Two of the reviewers were relatively positive, acknowledging the interesting and novel motivation of the paper, with potential practical applications.

The main concerns have to do with the assumptions and the fact that the dataset is synthetic. After looking at the paper myself, I believe said assumptions/dataset are in line with a large amount of the strategic classification literature, and that these issues are not paper-breaking.

As such, I recommend the paper to be accepted, however with low confidence and with the need for a SAC discussion. The meta-review also accounts for the fact this paper only received 3 out of 4 requested reviews.